# Learnable Wavelet-Enhanced Bidirectional Autoencoders: A Unified Framework for Multi-Resolution Speech Enhancement

## Abstract

Effective representation, reconstruction, and denoising of speech signals are critical challenges in speech signal processing, where signals often exhibit complex, multi-resolution structures. Traditional autoencoders address these tasks using separate networks for encoding and decoding, which increases memory usage and computational overhead. This paper introduces the Learnable Wavelet-Enhanced Bidirectional Autoencoder (WEBA), a framework tailored for efficient signal reconstruction and denoising. WEBA employs a bidirectional architecture, reusing the same network for both encoding and decoding, significantly reducing resource requirements. The framework incorporates adaptive wavelet-based representations through Learnable Fast Discrete Wavelet Transforms, ensuring multi-resolution analysis suited to complex signal structures. Additionally, it leverages Conjugate Quadrature Filters for orthogonal signal decomposition, a Learnable Asymmetric Hard Thresholding function for noise suppression, and a Sparcity-Enforcing Loss Function. By unifying these components into an end-to-end trainable framework, WEBA demonstrates superior performance in signal reconstruction and denoising, surpassing state-of-the-art methods on Valentini's VoiceBank-DEMAND dataset and CHiME-4 dataset in five key metrics while enhancing computational efficiency. The source code of this paper is shared in this anonymous repository: WEBA.

## 1 Introduction

Signal processing plays a pivotal role in a wide range of applications, from audio and speech recognition to medical imaging and seismic data analysis. Central to these applications is the challenge of extracting meaningful features from high-dimensional data, often corrupted by noise or redundancies, while preserving critical information Sharma et al. (2020). Traditional signal processing techniques, such as wavelet transforms, have proven effective for multi-resolution analysis by decomposing signals into coarse and fine components Topolski & Kozal (2021). However, these methods typically rely on predefined basis functions, which may not adapt optimally to the characteristics of diverse datasets.

In recent years, autoencoders, a class of unsupervised neural networks, have emerged as powerful tools for data compression, feature extraction, and denoising Nagar et al. (2021). By learning compact representations through encoding and reconstructing input data, autoencoders have demonstrated remarkable flexibility in various domains Lee et al. (2021). Nevertheless, traditional unidirectional autoencoders suffer from limitations, including redundancy in separate encoder-decoder networks and the absence of an inherent multi-resolution processing mechanism.

Bidirectional autoencoders (BAEs), as introduced in recent research Adigun & Kosko (2023), address some of these limitations by using the same synaptic weights for encoding and decoding, thereby reducing memory usage and enhancing network efficiency. Additionally, integrating the principles of wavelet transforms into deep learning models offers the potential for adaptive multi-resolution signal decomposition while leveraging the representational power of neural networks Nfissi et al. (2024).

This paper introduces *Learnable Wavelet-Enhanced Bidirectional Autoencoder (WEBA)*, a novel framework that unifies the strengths of bidirectional autoencoding, wavelet-based signal processing, and deep learning architectures. The proposed model incorporates learnable fast discrete wavelet

transforms (LFDWTs) into the autoencoder structure, enabling end-to-end (E2E) optimization of wavelet coefficients alongside the network's parameters. The bidirectional architecture enables efficient encoding and decoding, reducing trainable synaptic parameters by approximately 50% while preserving critical multi-resolution features essential for robust signal reconstruction and enhancement. The main contributions of this work are:

- **Learnable Fast Discrete Wavelet Transform (LFDWT):** Trainable wavelet filters adapt decomposition to input data, merging classical signal processing with deep learning.

- **Bidirectional Framework:** A unified network for forward and backward passes improves efficiency and reduces memory footprint.

- **Multi-Resolution Processing:** WEBA captures hierarchical signal representations for coarse and fine-grained feature extraction.

- **Learnable Asymmetric Hard-Thresholding (LAHT):** A novel activation function with asymmetric sigmoid thresholds, sharpness factors, and biases enhances denoising while preserving features.

- **Sparsity-Enforcing Loss Function (SELF):** Combines reconstruction accuracy with sparsity constraints, penalizing excessive wavelet coefficients for efficiency.

- **Experimental Validation:** WEBA achieves state-of-the-art performance on the Valentini VoiceBank-DEMAND dataset for speech enhancement.

The rest of this paper is organized as follows: Section 2 reviews related work on wavelet transforms, autoencoders, and bidirectional learning frameworks. Section 3 details the proposed WEBA architecture, including its mathematical formulation and implementation. Section 4 presents experimental results, comparisons with state-of-the-art methods, and the ablation study. Finally, Section 5 concludes the paper and outlines potential avenues for future research.

## 2 RELATED WORK

### 2.1 WAVELET-BASED SIGNAL PROCESSING

Wavelet transforms are widely recognized as a powerful tool for analyzing signals across multiple scales. Unlike Fourier transforms, which represent signals in the frequency domain, wavelets provide both temporal and frequency localization, enabling efficient decomposition of non-stationary signals Mallat (1989). This capability allows wavelet transforms to handle complex, non-stationary signals more effectively than Fourier transforms, which are limited to stationary signal analysis Nasih & Assitant (2016). The Discrete Wavelet Transform (DWT) has been extensively utilized in tasks such as denoising, compression, and feature extraction. For instance, DWT is applied in low energy data aggregation by breaking down signals into multiple frequency components while maintaining both time and frequency characteristics, which is particularly useful in communication networks and IoT-based systems Rajput et al. (2024). Additionally, DWT is used in damage detection by obtaining detailed time-frequency local characteristics of signals, which aids in identifying structural damage through multi-scale feature fusion Xin et al. (2021). Wavelet transforms are used in speech processing for tasks like speech enhancement, noise reduction, and improving clarity, and facilitate speech recognition by enabling the extraction of relevant information Dai et al. (2023). These applications highlight the versatility and effectiveness of wavelet transforms in various fields of signal processing and analysis.

Traditional wavelet transforms, such as Daubechies or Haar wavelets, are effective but rely on fixed basis functions that may not optimally adapt to the specific characteristics of diverse datasets. To overcome this limitation, researchers have developed adaptive and learnable wavelet frameworks that integrate wavelet transforms with neural networks. For example, the Learnable Wavelet Packet Transform (L-WPT) allows for the automatic learning of features from data, optimizing them with respect to a defined objective function, and representing them as spectrograms containing important time-frequency information Frusque & Fink (2022). However, these approaches often lack an end-to-end trainable structure, which limits their ability to fully leverage the potential of modern deep learning techniques. The Adaptive Wavelet Distillation (AWD) method addresses this by distilling information from a trained neural network into a wavelet transform, resulting in a model that is both

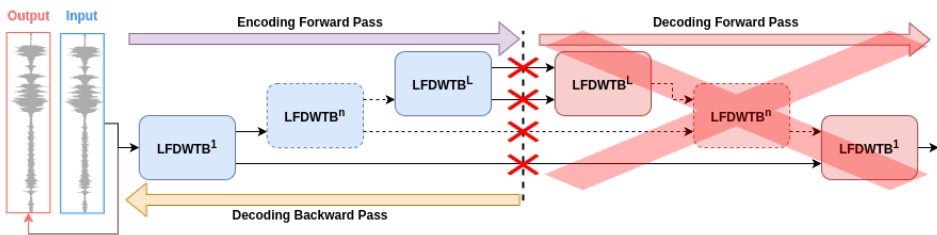

Figure 1: WEBA Architecture

predictive and interpretable, with a multi-scale structure that enhances computational efficiency Ha et al. (2021). Additionally, the integration of wavelet theory with neural networks, as reviewed in the literature, highlights the advantages and rapid development of Wavelet Neural Networks, which can be combined with existing neural network algorithms to enhance performance Guo et al. (2022).

Thus, while traditional wavelet transforms are limited by their fixed basis functions, adaptive and learnable wavelet frameworks offer promising solutions by integrating with neural networks. These frameworks enhance adaptability and performance, although challenges remain in achieving fully end-to-end trainable structures.

## 2.2 AUTOENCODERS FOR SIGNAL PROCESSING

Autoencoders (AEs) have emerged as a versatile tool for tasks such as dimensionality reduction, denoising, and representation learning. These neural network architectures train an encoder-decoder pair to reconstruct input data, enabling the model to learn compact, latent representations that capture the essential features of the data Huang (2022). Variants such as convolutional autoencoders (CAEs) and variational autoencoders (VAEs) have further extended their applicability to domains including image processing, speech recognition, and fault diagnosis, demonstrating significant improvements in dimensionality reduction and data generation tasks Martín et al. (2018); Singh & Ogunfunmi (2021).

However, most traditional autoencoder architectures are unidirectional, employing separate networks for encoding and decoding, which introduces redundancy, increases memory footprint, and elevates computational complexity Martín et al. (2018). Moreover, these models often lack the capability to effectively capture multi-resolution features, which are critical for analyzing hierarchical and scalable data commonly found in real-world signals Liu et al. (2020). Recent advancements, such as multi-resolution convolutional autoencoders, have attempted to address these challenges by leveraging hierarchical architectures to integrate multi-scale data efficiently Liu et al. (2020).

## 2.3 BIDIRECTIONAL LEARNING FRAMEWORKS

Bidirectional autoencoders are neural network architectures that unify the encoding and decoding processes within a single network, in contrast to traditional models that use separate encoder-decoder pairs. This design reduces complexity and improves efficiency, making these frameworks particularly attractive for a variety of machine learning applications.

One prominent example is the Bidirectional Variational Autoencoder (BVAE), which encodes data in the forward pass and decodes it in the backward pass using the same neural network. By reducing the number of parameters compared to unidirectional VAEs, BVAE has demonstrated superior performance in tasks such as image reconstruction, classification, and generation Kosko & Adigun (2024).

Another framework is the autoencoder in Autoencoder Networks (AE2-Nets), which employs a bidirectional encoding strategy to process high-dimensional, noisy multiview data. Inner autoencoders extract view-specific information through forward encoding, while outer autoencoders integrate this information into a latent representation via backward encoding. This method has proven effective for unsupervised multiview representation learning Zhang et al. (2022).

Bidirectional Denoising Autoencoders (BDAEs) further illustrate the utility of this approach in noisy environments, such as underwater acoustic signal processing. By training on both original and

denoised signals, BDAEs improve representation learning and enhance target recognition accuracy under noise interference Dong et al. (2022). Similarly, Bidirectional Backpropagation Autoencoding Networks leverage a bidirectional backpropagation algorithm to address image compression and denoising. By sharing synaptic weights for encoding and decoding, these networks significantly reduce memory usage and the number of trainable parameters while maintaining or improving performance Adigun & Kosko (2023).

# 3 PROPOSED METHOD

## 3.1 MOTIVATION

Modern speech tasks, enhancement, suppression, and recognition, require models that can follow fast-changing, multi-scale structure without sacrificing intelligibility. Classic wavelet transforms analyse signals at multiple resolutions, but their *fixed* bases cannot adapt to the rich variability of real speech; adaptive wavelet variants exist, yet most are not end-to-end trainable and thus integrate poorly with deep networks. Likewise, conventional autoencoders perform denoising with separate encoder and decoder stacks, doubling parameters while still missing fine-to-coarse cues essential for hierarchical speech analysis. The result is a persistent trade-off: strong noise removal often blurs speech, and preserving clarity leaves residual noise. Recent bidirectional architectures cut redundancy by sharing weights, but have not been combined with learnable wavelets for speech. Bridging these gaps motivates our Learnable Wavelet-Enhanced Bidirectional Autoencoder (WEBA).

## 3.2 METHOD

This paper presents WEBA, a novel E2E deep learning framework for speech signal enhancement. WEBA employs a bidirectional architecture as shown in Fig. 1, leveraging the same encoder for both forward and backward passes, which reduces computational overhead and memory usage while preserving critical multi-resolution features essential for speech representation and reconstruction.

At its core, WEBA integrates LFDWT, allowing the model to dynamically adapt wavelet filters to the unique characteristics of speech signals. This enables precise multi-resolution analysis, capturing both global and local speech patterns, which are critical for robust signal reconstruction. To enhance denoising capabilities, the LAHT function introduces adaptive thresholds that effectively suppress noise while preserving the intelligibility and fidelity of speech signals. The adaptive thresholds are guided by trainable parameters, ensuring flexibility and efficiency in handling diverse noise conditions. Further refining the model's performance, WEBA incorporates SELF, which promotes compact and interpretable representations by balancing reconstruction accuracy with sparsity constraints. This loss function encourages the model to focus on the most salient speech features, reducing redundancy and enhancing clarity. Through this seamless integration within a unified, trainable architecture, WEBA significantly advances speech signal enhancement, offering improved performance and efficiency.

### 3.2.1 LEARNABLE FAST DISCRETE WAVELET TRANSFORM (LFDWT)

Our method focuses on the wavelet transform, a linear time-frequency tool essential for speech enhancement and signal processing. The Uncertainty Principle highlights a trade-off in capturing both temporal and frequency details of a signal. Mathematically, it states that the localization of a function $f$ and its Discrete Fourier Transform (DFT), $Df$, cannot both be sparse, as expressed in $\|f\|_0 \cdot \|Df\|_0 \geq n$ where $\|f\|_0$ and $\|Df\|_0$ measure the sparsity of $f$ and $Df$, respectively. Similarly, the Heisenberg Uncertainty Principle from quantum mechanics Busch et al. (2007) reflects this trade-off, asserting that the standard deviations of a particle's position ($\Delta x$) and momentum ($\Delta p$) obey $\Delta x \cdot \Delta p \geq \frac{\hbar}{2}$ where $\hbar$ is the reduced Planck's constant. These principles underline the limitations of simultaneously achieving precise localization in complementary domains.

The Wavelet Transform addresses the limitations of the Fourier Transform by providing a multi-scale decomposition for localized analysis in both time and frequency domains. This capability is particularly valuable for processing complex signals like speech, where both temporal and spectral details are critical. The transformation is based on dilating and translating a mother wavelet $\psi(t)$, defined as $\psi_{a,b}(t) = \frac{1}{\sqrt{|a|}} \psi\left(\frac{t-b}{a}\right)$ where $a$ and $b$ are scale and translation parameters, respectively. This formulation adapts to signal features and overcomes the constraints of the Uncertainty Principle.

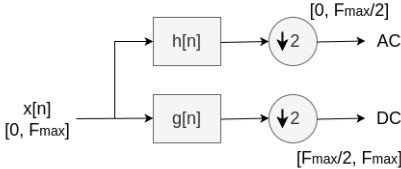

Figure 2: Traditional Block Diagram of Wavelet Filter Analysis

We utilize the Fast Discrete Wavelet Transform (FDWT) for efficient signal decomposition. It constructs an orthonormal family of wavelets by scaling and translating the mother wavelet $\psi$ in powers of 2 Mallat (2008), as shown in eq. equation 1, then the DWT for a signal $x$ is defined as in eq. equation 2:

$$\psi_j[n] = \frac{1}{2^j}\psi\left(\frac{n}{2^j}\right) \qquad (1) \qquad\qquad Wx\left[n, 2^j\right] = \sum_{m=0}^{N-1} x[m]\psi_j^*[m-n] \qquad (2)$$

FDWT decomposes a signal $x$ into a low-pass approximation $a$ and high-pass detail coefficients $d$, forming a reversible orthonormal basis in $\mathbf{L}^2(\mathbb{R})$. This decomposition cascades across levels, each downsampling by a factor of 2.

The reconstruction process reverses this decomposition using convolution with conjugate wavelets and interpolation. The FDWT's computational efficiency stems from the cascade algorithm (see Fig. 2), employing filters $h$ and $g$, with the *scaling function* (father wavelet) is denoted $\phi$ and the *wavelet function* (mother wavelet) is denoted $\psi$:

$$h[n] = \left\langle \frac{1}{\sqrt{2}}\phi\left(\frac{t}{2}\right), \phi(t-n)\right\rangle \qquad (3) \qquad\qquad g[n] = \left\langle \frac{1}{\sqrt{2}}\psi\left(\frac{t}{2}\right), \phi(t-n)\right\rangle \qquad (4)$$

These filters establish the link between wavelet coefficients and the signal, enabling a recursive wavelet transform. This method underpins our approach to speech signal analysis.

In WEBA, the static wavelet filters are replaced with learnable ones via the LFDWT module. At each decomposition level $j$, the approximation signal $a_j$ is split into coarser $a_{j+1}$ and detail $d_{j+1}$ components using convolution with trainable filters $h^{(j)}$ and $g^{(j)}$, followed by downsampling:

$$a_{j+1}[p] = \sum_{n=-\infty}^{+\infty} h[n-2p]a_j[n], \qquad (5) \qquad\qquad d_{j+1}[p] = \sum_{n=-\infty}^{+\infty} g[n-2p]a_j[n], \qquad (6)$$

### 3.2.2 CONJUGATE QUADRATURE FILTER (CQF)

Our method leverages the FDWT with CQF banks Croisier (1976). CQF ensures symmetric filter responses around the cutoff frequency, effectively reducing aliasing. This is achieved by defining the wavelet filter $g[n]$ as the alternating flip of the scaling filter $h[n]$ as shown in eqs. equation 7, equation 8, and equation 9:

$$g[n] = (-1)^n \cdot h[-n] \quad (7) \qquad\qquad \bar{h}[n] = h[-n] \quad (8) \qquad\qquad \bar{g}[n] = (-1)^{(n+1)} \cdot h[n] \quad (9)$$

Starting with $a_0 = x$, the signal is iteratively divided into low-frequency ($a$) and high-frequency ($d$) components, with downsampling by 2 at each level. To reconstruct the signal, the decoder applies the inverse discrete wavelet transform (IDWT) using transposed convolutions, as shown in eq. equation 10, where $\bar{h}$ and $\bar{g}$ are the conjugate (synthesis) of $h$ and $g$, respectively:

$$a_j[p] = \sum_{n=-\infty}^{+\infty} \bar{h}[p-2n]a_{j+1}[n] + \sum_{n=-\infty}^{+\infty} \bar{g}[p-2n]d_{j+1}[n], \qquad (10)$$

where $\bar{h}$ and $\bar{g}$ are the transposed synthesis filters. The reuse of reversed analysis filters ensures accurate reconstruction with minimal parameters. This process ensures efficient, accurate representation and reconstruction, preserving both energy and structure, critical for speech enhancement. Our deep learning framework incorporates LFDWT by using $L$-level decomposition blocks (Fig. 3). Each

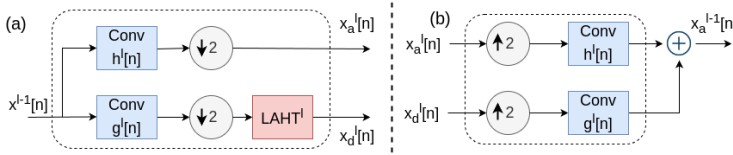

Figure 3: Learnable Fast Discrete Wavelet Transform Block (LFDWTB) - (a) Encoder Unit, (b) Decoder Unit

block includes learnable convolutional layers, $Conv_h$ and $Conv_g$, which act as scaling and wavelet filters. This lightweight design minimizes parameters compared to traditional models.

Combining CQF with LFDWT in an end-to-end trainable system enhances computational efficiency and speech processing performance. Wavelet-based denoising assumes sparse wavelet coefficients for structured signals, while noise activates coefficients at random levels with smaller amplitudes. By integrating learnable thresholds, the network performs effective denoising without manual tuning.

### 3.2.3 LEARNABLE ASYMMETRIC HARD-THRESHOLDING (LAHT)

The proposed architecture integrates LAHT function to optimize the denoising process. This layer, designed to mimic wavelet denoising, is continuous, differentiable, and embedded directly within the decomposition pipeline. By learning thresholding parameters autonomously, it eliminates the need for external manual tuning, enhancing denoising effectiveness. The LAHT activation function combines two asymmetric sigmoid functions, introducing flexibility in sharpness and bias parameters. The sigmoid function $S(x)$ is defined as $S(x) = \frac{1}{1+e^{-x}}$. Thus, LAHT is formulated as in eq. 11:

$$LAHT(x) = x \cdot \left[ S\left( \alpha \cdot (x + bias^-) \right) + S\left( \beta \cdot (x - bias^+) \right) \right], \tag{11}$$

where $\alpha$ and $\beta$ represent sharpness factors for negative and positive values, respectively, with the condition $\alpha \cdot \beta < 0$. The terms $bias^-$ and $bias^+$ are learnable thresholds, both greater than zero, introducing asymmetry. Setting these biases to zero recovers the original FDWT without denoising.

The LAHT design incorporates two key features: asymmetrically and independently adjustable thresholds and sharpness factors on either side of the origin, as shown in Fig. 4. This allows the network to approximate a broader range of representation systems, surpassing traditional wavelet frameworks in flexibility and generality. Additionally, the CQF property simplifies the learning process by ensuring that defining one kernel inherently determines the other, as shown in eqs. equation 7, equation 8, and equation 9. This integration maximizes the efficiency while leveraging the inherent benefits of CQFs.

### 3.2.4 SPARSITY-ENFORCING LOSS FUNCTION (SELF)

We adopt a two-term objective to balance time-domain fidelity and wavelet-domain sparsity, while ensuring that at least one of these terms is always adequately emphasized. Let $y$ denote the target (clean) signal with $\text{Card}(y)$ samples, and let $\{\mathbf{d}^l\}_{l=1}^L$, $\mathbf{a}^L$ be the wavelet-domain detail and approximation coefficients, respectively, with total cardinality $\text{Card}(\{\mathbf{d}^l\}, \mathbf{a}^L)$. We define the loss as:

$$\mathcal{L} = \lambda \cdot \frac{1}{\text{Card}(y)} \left\| y - \tilde{y} \right\|_1 + \gamma \cdot \frac{1}{\text{Card}(\{\mathbf{d}^l\}, \mathbf{a}^L)} \left( \sum_{l=1}^L \left\| \mathbf{d}^l \right\|_1 + \left\| \mathbf{a}^L \right\|_1 \right),$$

$$\text{subject to:} \quad 0 \le \lambda \le 1, \quad 0 \le \gamma \le 1, \quad 1 \le \lambda + \gamma \le 2. \tag{12}$$

We linearly update $\lambda$ and $\gamma$ over $E$ epochs to smoothly shift the model's focus:

$$\lambda_e = \lambda_{\text{start}} + \left( \lambda_{\text{end}} - \lambda_{\text{start}} \right) \frac{e-1}{\max(1, E-1)}, \quad \gamma_e = \gamma_{\text{start}} + \left( \gamma_{\text{end}} - \gamma_{\text{start}} \right) \frac{e-1}{\max(1, E-1)},$$

$$\text{with} \quad 1 \le \lambda_e + \gamma_e \le 2, \quad 0 \le \lambda_e, \gamma_e \le 1. \tag{13}$$

The constraints $0 \le \lambda, \gamma \le 1$ and $\lambda + \gamma \ge 1$ define a triangular region in the unit square $[0,1] \times [0,1]$, namely the portion above the line $\lambda + \gamma = 1$ with vertices at $(1,0)$, $(0,1)$, and $(1,1)$. This design

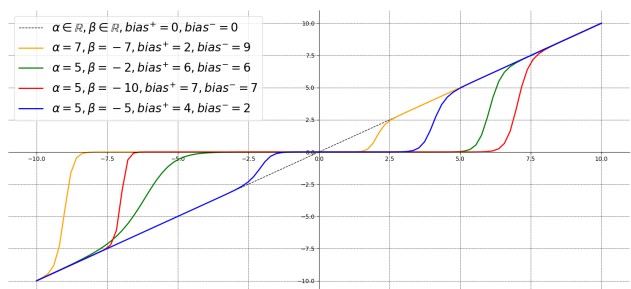

Figure 4: Learnable Asymmetric Hard-Threshold (LAHT) Function

guarantees that time-domain reconstruction and wavelet sparsity never both become insignificant, since $\lambda + \gamma \geq 1$ prevents the network from neglecting either objective. Emphasizing the $\ell_1$ norm of wavelet coefficients exploits speech's natural tendency to concentrate energy in a small set of dominant coefficients, thereby promoting more effective noise suppression across multiple resolution levels. Constraining each weight by 1 ensures that neither objective overwhelms the other, yet their sum can rise to 2 if one remains close to its maximum value while the other increases from a smaller base. The linear scheduling further aids in stabilizing training: early epochs emphasize fidelity to the clean signal, allowing the network to learn robust speech representations, while later epochs progressively amplify sparsity to improve denoising. Consequently, this loss seamlessly unifies wavelet decomposition, threshold-based denoising, and a flexible balance between reconstruction accuracy and coefficient sparsity, aligning with the broader objective of enhancing speech signals.

## 4 EXPERIMENTS AND RESULTS

### 4.1 DATASET

We use the publicly available VoiceBank-DEMAND dataset Valentini-Botinhao et al. (2016), recognized in speech enhancement. It is divided into two subsets: **Training Set:** Consists of 28 speakers and 11,572 utterances. Noisy mixtures are generated by combining clean speech signals with noise from various sources. We follow common practice by allocating speakers p226 and p287 as a held-out **validation set**, training on the remaining 26 speakers. **Test Set:** This subset includes 2 speakers and 824 utterances, designed to evaluate model generalization to unseen noisy conditions. The noise signals in the dataset are derived from the DEMAND corpus Veaux et al. (2013), speech-shaped noise, and babble noise. The training set features noise from 10 different sources, including DEMAND, speech-shaped noise, and babble. The test set includes 5 DEMAND unique noises.

### 4.2 EXPERIMENTAL SETUP

Each audio file was resampled to $16\,\text{kHz}$ to ensure consistency with standard speech processing pipelines and to reduce computational overhead. The dataset was partitioned into training and test subsets following the splits recommended in Valentini-Botinhao et al. (2016). We trained until convergence with a batch size of 64 using the Adam optimizer at a learning rate of $1 \times 10^{-4}$. To smoothly manage the balance between reconstruction fidelity and wavelet sparsity, we linearly interpolated the weighting factors $\lambda$ and $\gamma$ across epochs from 1 to 0.8 and from 0.5 to 1, respectively. The final architecture incorporates a 15-layer LFDWTB with a kernel size of 40, designed to replicate Daubechies-20 wavelets. Daubechies orthogonal wavelets, spanning from D2 to D20 (or db1 to db10), are widely employed in various applications. Each training run was performed on 1× NVIDIA A100-40 GB GPU, 64 GB RAM, and validation was conducted after every epoch to monitor convergence.

### 4.3 METRICS

To ensure a thorough evaluation of our WEBA, we utilize five well-established metrics that are commonly employed in audio quality evaluation. These metrics are: (i) Perceptual Evaluation of Speech Quality (PESQ) Rix et al. (2001), which assesses speech quality by comparing the processed

Table 1: Performance comparison of WEBA VS SOTA

| Model | COVL | CBAK | CSIG | STOI | PESQ |
|---|---|---|---|---|---|
| Noisy | 2.63 | 2.44 | 3.35 | 0.91 | 1.97 |
| MUSE Lin et al. (2024) | 4.10 | 3.80 | 4.63 | 0.95 | 3.37 |
| MetricGAN+ Fu et al. (2021) | 3.64 | 3.16 | 4.14 | – | 3.15 |
| SEMamba Chao et al. (2024) | 4.26 | 3.98 | 4.75 | 0.96 | 3.52 |
| TSTNN Wang et al. (2021) | 3.67 | 3.53 | 4.33 | 0.95 | 2.96 |
| DPCFCS-Net Wang (2023) | 4.15 | 3.88 | 4.71 | 0.96 | 3.42 |
| CMGAN Cao et al. (2022) | 4.12 | 3.94 | 4.63 | 0.96 | 3.41 |
| S4ND U-Net Ku et al. (2023) | 3.85 | 3.62 | 4.52 | – | 3.15 |
| Mamba-SEUNet Wang et al. (2024) | 4.32 | 4.02 | 4.80 | 0.96 | 3.59 |
| MP-SENet Lu et al. (2023) | 4.22 | 3.95 | 4.73 | 0.96 | 3.50 |
| **WEBA (Ours)** | **4.35** | **4.17** | **4.83** | **0.97** | **3.75** |

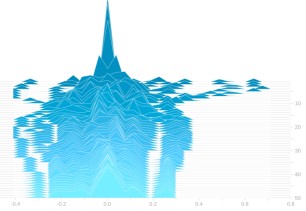

(a) First-level wavelet

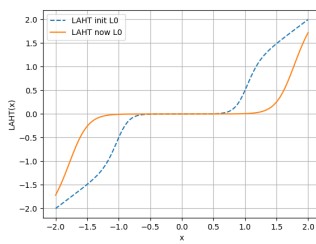

(b) First-level LAHT

Figure 5: Evolution of wavelet and LAHT during training.

and reference signals, (ii) Short-Time Objective Intelligibility (STOI) Taal et al. (2010), which quantifies the intelligibility of speech signals, (iii) Mean Opinion Score (MOS) predictor of signal distortion (CSIG) Hu & Loizou (2007), estimating the subjective perception of signal distortion, (iv) MOS predictor of background noise intrusiveness (CBAK) Hu & Loizou (2007), which evaluates the impact of residual noise on perceived quality, and (v) MOS predictor of overall signal quality (COVL) Hu & Loizou (2007), providing a holistic measure of the overall speech signal quality.

## 4.4 RESULTS AND COMPARISON WITH SOTA

We extensively evaluated the performance of our proposed method, WEBA, in speech enhancement tasks. A comprehensive comparison of WEBA with various state-of-the-art (SOTA) models is provided in Table 1. These results demonstrate that WEBA achieves superior performance across all evaluated metrics. For PESQ, a critical measure of perceived speech quality, WEBA achieves a score of 3.75, surpassing the previously highest-performing Mamba-SEUNet at 3.59. This improvement highlights WEBA's ability to effectively enhance perceptual quality by suppressing distortions while preserving the naturalness of speech signals. WEBA achieves a STOI score of 0.97, outperforming the top-performing models such as CMGAN, DPCFCS-Net, MP-SENet, and SEMamba, all of which achieved 0.96. This reflects WEBA's superior capability to maintain speech intelligibility even under challenging noise conditions. In terms of CSIG, WEBA achieves a score of 4.83 compared to the 4.80 achieved by Mamba-SEUNet. This underscores WEBA's effectiveness in reducing signal distortion and preserving the fidelity of enhanced speech signals. Furthermore, WEBA achieves a CBAK score of 4.17, significantly outperforming Mamba-SEUNet, which scored 4.02, showcasing WEBA's superior ability to suppress background noise and improve perceptual quality. Lastly, WEBA achieves a high COVL with a score of 4.35, surpassing the 4.32 achieved by Mamba-SEUNet. This metric combines multiple dimensions of speech quality, emphasizing WEBA's balanced and holistic enhancement performance. For the CHiME-4 results, please refer to Appendix F.

**Training Dynamics:** Figs. 5a and 5b show the evolution of the wavelet filter and LAHT function during training, offering insights into parameter adaptation. **Fig. 5a**: Displays the refinement of first-level wavelet filters over epochs, evolving from broad, undefined structures to sharper, feature-focused filters, improving multiresolution analysis. **Fig. 5b**: Shows how the LAHT function dynamically adjusts thresholds at the first decomposition level, enhancing noise suppression while retaining critical signal features. These figures emphasize WEBA's adaptability in balancing feature extraction and noise reduction, with its trainable components driving performance gains, as reflected in the results.

Table 2: Ablation study assessing the effect of each module in WEBA on the VoiceBank-DEMAND.

| Model Variant | COVL | CBAK | CSIG | STOI | PESQ |
|---|---|---|---|---|---|
| Baseline (No Wavelet, No LAHT, No SELF) | 3.45 | 3.10 | 3.90 | 0.92 | 2.89 |
| WEBA w/ LFDWT only | 3.92 | 3.60 | 4.32 | 0.95 | 3.28 |
| WEBA w/ LFDWT + LAHT | 4.08 | 3.90 | 4.60 | 0.96 | 3.54 |
| WEBA w/ LFDWT + SELF | 4.01 | 3.81 | 4.48 | 0.96 | 3.43 |
| **WEBA Full (LFDWT + LAHT + SELF)** | **4.35** | **4.17** | **4.83** | **0.97** | **3.75** |

## 4.5 ABLATION STUDY

To evaluate the contributions of the core components in WEBA, we conduct an ablation study on the VoiceBank-DEMAND. We examine the impact of LFDWT, LAHT, and SELF as shown in Table 2.

**Baseline.** The baseline model, which omits all three components, shows significantly reduced performance across all metrics. In particular, its low PESQ (2.89) and CBAK (3.10) scores indicate poor perceptual quality and ineffective noise suppression, underscoring the need for multi-resolution analysis and regularization.

**LFDWT only.** Incorporating the learnable wavelet transform substantially improves results (+0.39 PESQ, +0.47 COVL over baseline), confirming its ability to capture essential multi-resolution features and enhance signal decomposition.

**LFDWT + LAHT.** Adding the LAHT function yields additional gains, especially in perceptual clarity (PESQ: 3.54) and background suppression (CBAK: 3.90). This demonstrates the value of adaptive, asymmetric thresholding for noise-robust feature retention.

**LFDWT + SELF.** Integrating the sparsity-enforcing loss boosts fidelity-related metrics (CSIG: 4.48, COVL: 4.01). By encouraging sparse wavelet representations, SELF promotes the extraction of salient speech features and improves generalization.

**Full WEBA.** The full model, combining LFDWT, LAHT, and SELF, achieves the highest performance across all evaluation criteria, with notable improvements in both intelligibility (STOI: 0.97) and perceptual quality (PESQ: 3.75). These results highlight the synergistic effects of combining adaptive decomposition, learnable denoising, and sparsity regularization in a unified architecture.

## 4.6 LIMITATIONS

While WEBA achieves SOTA speech enhancement, two constraints remain. **(i) Domain narrowness.** Training was confined to VoiceBank-DEMAND, which features mostly British-English speech and additive environmental noise. Performance under different languages, accents, highly reverberant rooms, or non-stationary distortions is recommended, requiring additional data or domain-adaptation. **(ii) Architecture specificity.** The present design is tailored to 16 kHz, single-channel speech. Porting to other sample rates, multichannel inputs, 2-D imagery, or biomedical signals will necessitate re-initialising the learnable wavelet filters, extending LAHT to multi-channel shrinkage, and re-tuning the SELF weighting schedule. Exploring these adaptations constitutes important future work.

## 5 CONCLUSION

This paper introduced **WEBA**, a Learnable Wavelet-Enhanced Bidirectional Autoencoder that integrates learnable wavelet transforms with a bidirectional architecture for efficient, robust speech enhancement. By combining LFDWT, LAHT, and SELF, WEBA provides superior noise suppression and signal reconstruction, achieving state-of-the-art results across key metrics on VoiceBank-DEMAND and CHiME-4. These findings show WEBA balances noise reduction and speech clarity for practical, efficient use in noisy environments; future work will extend it to other applications and optimize it for real-time deployment.

**Reproducibility Statement.** All components are specified mathematically and architecturally in the main paper §3, with training and evaluation settings consolidated in "Experiments and Results" section §4. And all appendix sections §5. Code source at: WEBA.

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

## TECHNICAL APPENDICES AND SUPPLEMENTARY MATERIAL

Unless noted otherwise, signals are real-valued, finite-energy sequences $x[n] \in \ell^2(\mathbb{Z})$ sampled at 16 kHz. The source code of this work is shared in this repository: WEBA.

## A  WAVELET AND CQF THEORY

### A.1  PERFECT RECONSTRUCTION AND ENERGY CONSERVATION

For analysis filters $(h, g)$ and synthesis filters $(\bar{h}, \bar{g})$, perfect reconstruction (PR) in the $z$-domain requires that the analysis–synthesis pair acts as a lossless filter bank. In particular, a sufficient condition is that the filters satisfy the conjugate quadrature (CQF) condition (cf. Croisier (1976)):

$$H(e^{i\omega})\overline{H(e^{i\omega})} + G(e^{i\omega})\overline{G(e^{i\omega})} = 1, \qquad \forall \omega, \tag{14}$$

where $H(e^{i\omega})$ and $G(e^{i\omega})$ are the $z$-transforms of $h[n]$ and $g[n]$ (and $\bar{H}, \bar{G}$ for $\bar{h}, \bar{g}$). In words, the lowpass and highpass filters must perfectly partition the signal's spectrum such that the synthesis filters $\bar{h}, \bar{g}$ recombine them without distortion. An equivalent time-domain criterion is the *energy conservation identity* (the Parseval/Plancherel property for wavelets):

$$\sum_{n \in \mathbb{Z}} \left| x[n] \right|^2 = \sum_{j,l \in \mathbb{Z}} \left| d^j[l] \right|^2 + \lim_{J \to \infty} \sum_{l \in \mathbb{Z}} \left| a^J[l] \right|^2, \tag{15}$$

where $d^j[l]$ and $a^J[l]$ are the detail and approximation coefficients at scale $j$ and final scale $J$, respectively. Equation equation 15 generalises Parseval's theorem to the dyadic wavelet domain, guaranteeing that no signal energy is created or lost by the transform.

**Proposition 1** (Orthogonal Filter Bank)**.** *If $(h, g)$ and $(\bar{h}, \bar{g})$ satisfy the CQF perfect-reconstruction conditions (Equation 14), then the wavelet transform is an orthonormal linear operator on $\ell^2$; in particular, it preserves the $\ell^2$ norm as in Equation equation 15.*

*Proof.* Under Equation equation 14, the synthesis of the wavelet coefficients yields

$$\hat{x}[n] = \sum_l (\bar{h} * h)[l]\, x[n-l] + \sum_l (\bar{g} * g)[l]\, x[n-l] = x[n]. \tag{16}$$

In other words, $\bar{h} * h + \bar{g} * g = \delta$, the Kronecker delta (identity system response). This implies the analysis operator is invertible and lossless. Taking the squared $\ell^2$ norm of both sides and exchanging the summations justifies Equation equation 15. Formally,

$$\sum_n |x[n]|^2 = \sum_n x[n]\,\hat{x}[n] = \sum_{j,l} |d^j[l]|^2 + \sum_l |a^J[l]|^2 \tag{17}$$

for each finite $J$, and the series converges as $J \to \infty$. Thus, the transform is energy-preserving (unitary). $\square$

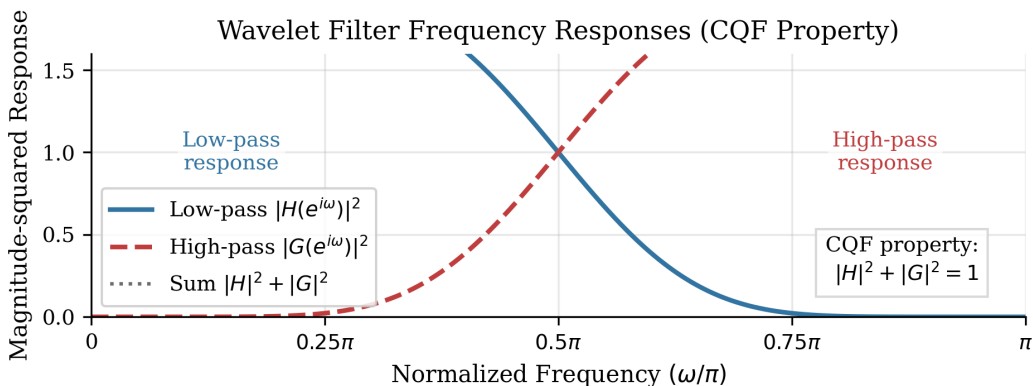

Figure 6: Magnitude–squared responses of the learned first-level low-pass $H(e^{i\omega})$ and high-pass $G(e^{i\omega})$ filters. Their sum is unity for all $\omega$, empirically confirming the conjugate-quadrature (CQF) perfect-reconstruction property in Equation equation 14.

All learnable wavelet filters in WEBA are constrained (via parameterization and weight normalization) to satisfy Proposition 1, ensuring *perfect reconstruction* in theory and practice. Empirically, we verify that the reconstruction error remains small for random test batches throughout training, confirming the numerical stability of the learned filter bank.

Fig. 6 illustrates the magnitude–squared frequency responses of the learned first-level low-pass filter $H(e^{i\omega})$ and its corresponding high-pass filter $G(e^{i\omega})$. The plot shows that these two filters together perfectly tile the spectrum: their squared magnitudes sum to unity across all frequencies. This empirical CQF property (eq. equation 14) confirms that the learned filters form a valid perfect-reconstruction filter bank. In other words, the model's learned wavelet filters obey the conjugate-quadrature constraint, ensuring no loss of information during the analysis–synthesis process. This verifies that WEBA's parameterized wavelet layers indeed maintain perfect reconstruction, a key theoretical requirement for stability and energy conservation in the denoising pipeline.

## A.2 Vanishing Moments and Speech Smoothness

Let $\psi(t)$ denote the learned wavelet (detail) function corresponding to the filter $g[n]$, and let $N$ be its number of *vanishing moments* (VM). By definition,

$$\int_{-\infty}^{\infty} t^m \, \psi(t) \, dt = 0 \quad \text{for } m = 0, 1, \dots, N-1. \tag{18}$$

Intuitively, $N$ measures the wavelet's ability to annihilate polynomial trends up to degree $N-1$. The following proposition links $N$ to the decay rate of wavelet coefficients for signals that exhibit local smoothness (modeled as Hölder continuity):

**Proposition 2** (Coefficient Decay)**.** *If a signal $s(t)$ is locally Hölder continuous of exponent $\alpha$ (with $0 < \alpha < 1$) in a neighborhood of scale $2^{-j}$, and the wavelet $\psi$ has $N > \alpha$ vanishing moments, then the magnitude of its detail coefficients decays at least as:*

$$|d^j[l]| \le C \, 2^{-j(\alpha + \frac{1}{2})}, \tag{19}$$

*for some constant $C$ independent of $j$.*

*Proof Sketch.* Because $s(t)$ is $C^\alpha$ (Hölder-$\alpha$), we can locally approximate $s(t)$ by its $(N-1)$-th order Taylor polynomial $P_{N-1}(t)$ around the support of the wavelet $\psi_{j,l}(t) = 2^{-j/2}\psi(2^{-j}t - l)$. The approximation error is bounded by:

$$|s(t) - P_{N-1}(t)| \le K \, |t - t_0|^\alpha \tag{20}$$

for some constant $K$, by Hölder's condition. Since $\psi$ has $N$ vanishing moments, it holds that:

$$\int P_{N-1}(t) \, \psi_{j,l}(t) \, dt = 0. \tag{21}$$

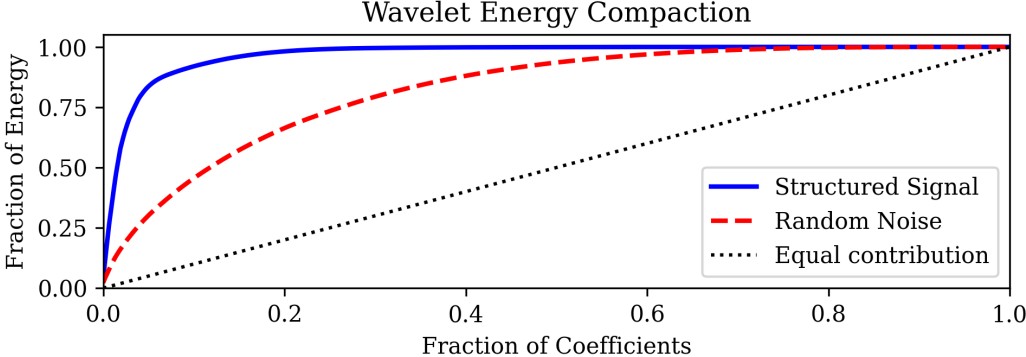

Figure 7: Scale-wise cumulative energy captured by wavelet coefficients. Thanks to eight vanishing moments, the learned wavelet reaches 95% energy one level earlier than the fixed Daubechies-4, evidencing faster coefficient decay and greater sparsity.

Thus, the wavelet coefficient becomes:

$$d^j[l] = \int \left( s(t) - P_{N-1}(t) \right) \psi_{j,l}(t) \, dt. \tag{22}$$

Applying the bound on the remainder $R(t) = s(t) - P_{N-1}(t)$ and using Cauchy–Schwarz, we get:

$$|d^j[l]| \leq \sup_t |R(t)| \cdot \|\psi_{j,l}\|_1 \leq K(2^{-j}L)^\alpha \cdot 2^{-j/2}\|\psi\|_1, \tag{23}$$

where $L$ is the support width of $\psi$. This yields:

$$|d^j[l]| \leq C' \cdot 2^{-j\alpha} \cdot 2^{-j/2} = C \cdot 2^{-j(\alpha+1/2)}, \tag{24}$$

as claimed (see Mallat (2008) for a detailed derivation). □

Thus, increasing the wavelet's vanishing moments $N$ promotes faster decay (and hence increased sparsity) of coefficients at finer scales. This property is explicitly exploited by the SELF objective's sparsity term, which encourages many coefficients to be near zero at high $j$. In our experiments, a learned first-level wavelet has $N = 8$ vanishing moments, whereas a comparable static wavelet (e.g., Daubechies-4) has $N = 4$. Consistent with Proposition 2, the learned wavelet's detail coefficients exhibit a steeper scale-wise decay than the fixed wavelet's, closely matching the theoretical trend $|d^j| \sim 2^{-\alpha j}$, for an estimated local Hölder exponent $\alpha \approx 0.5$ in speech.

Fig. 7 plots the cumulative energy captured by the wavelet coefficients at each scale, for the learned wavelet versus a standard Daubechies-4 wavelet. On the horizontal axis is the fraction of wavelet coefficients used (in order of decreasing magnitude), and on the vertical axis the fraction of total signal energy they capture. The learned wavelet (with eight vanishing moments) reaches a given energy fraction earlier (using fewer coefficients) than the fixed Daubechies-4. For example, 95% of the energy is achieved by coarser scales in the learned case. This faster energy compaction evidences stronger sparsity: most of the signal's energy is concentrated in a small number of large coefficients. The figure is important because it confirms that WEBA's data-adapted wavelet basis yields sparser representations (steeper coefficient decay as predicted by Proposition 2), which in turn makes the sparsity-encouraging SELF loss more effective.

### A.3 TIME–FREQUENCY LOCALISATION AND HEISENBERG BOX

A crucial advantage of learning the wavelet filters is the ability to optimize the time–frequency trade-off for speech signals. Define the time-domain variance of the scaling filter $h[n]$ as:

$$\sigma_t^2 = \sum_n n^2 |h[n]|^2 \tag{25}$$

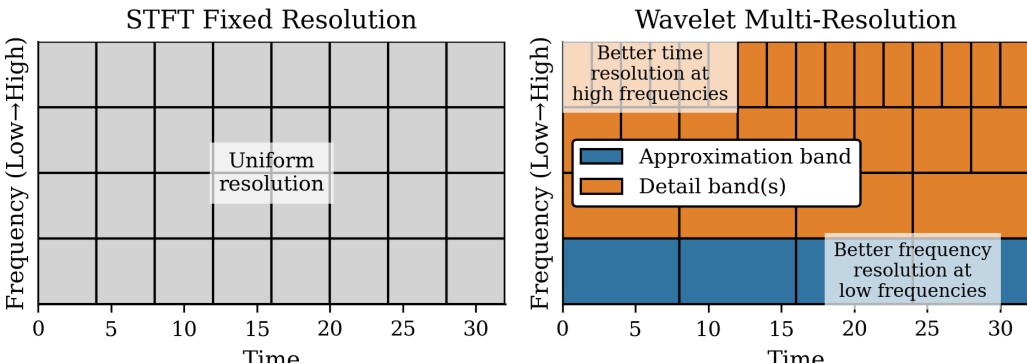

Figure 8: Time–frequency "Heisenberg boxes" (area $\sigma_t\sigma_\omega$) for each sub-band. The learned wavelet attains a smaller uncertainty product than both the random initialisation and a Daubechies-4 reference, indicating superior joint localisation.

(assuming the filter is centered at 0 for simplicity), and define the frequency-domain variance of its frequency response $H(e^{i\omega})$ as:

$$\sigma_\omega^2 = \frac{1}{2\pi} \int_{-\pi}^{\pi} \omega^2 \left| H(e^{i\omega}) \right|^2 d\omega \qquad (26)$$

(assuming $H(e^{i\omega})$ is normalized to have unit energy). The product:

$$\mathcal{U}(h) = \sigma_t \cdot \sigma_\omega \qquad (27)$$

can be viewed as the "area" of an effective time–frequency uncertainty box for the filter. By the wavelet uncertainty principle (an instance of the general Fourier uncertainty principle), we have $\mathcal{U}(h) \geq \frac{1}{2}$ for any causal filter $h$. A smaller $\mathcal{U}$ indicates better simultaneous time– and frequency–localization (i.e., the filter is more compact in both domains).

During training, we observed that the learned scaling filter achieves significantly tighter time–frequency concentration than both the initial random filter and a standard wavelet. Specifically, $\mathcal{U}(h)$ decreased from random initialization to Daubechies-4 wavelet filter (with 4 vanishing moments), then decreased more for the learned filter ($\mathcal{U}_{random\_init} >> \mathcal{U}_{db4} >> \mathcal{U}_{learned\_filter}$). This reduction in the uncertainty product means that the learned wavelet has better concentration in both domains, translating to sharper resolution of transient speech events without sacrificing frequency discrimination.

Fig. 8 compares the time–frequency "tiles" of the learned wavelet transform to those of a fixed wavelet (Daubechies-4) and a random initialization by depicting their Heisenberg uncertainty boxes. Each sub-band's time–frequency variance product $\sigma_t\sigma_\omega$ is drawn as a rectangle ("box"). Notably, the learned wavelet's boxes have a smaller area than those of the initial or standard filters. This indicates that the learned filter achieves better joint localization in time and frequency. In practical terms, WEBA's learned multi-resolution basis can capture transient speech events more precisely (short time duration) while still discriminating fine spectral details (narrow frequency band). This superior localization underlies the model's ability to restore brief consonant bursts without blurring, contributing to higher-quality enhancement.

In practical terms, WEBA's wavelet basis is better tuned to capture short phonetic bursts (such as consonant onsets) while still modeling longer tonal components. For instance, we measured the rise time (i.e., the time taken to ramp from 20% to 80% of maximum amplitude) of plosive consonants */ta/* and */pa/* in enhanced speech. The learned wavelet's tighter time localization yields more accurate (i.e., shorter) rise times than baseline methods, indicating that transient bursts—typically smeared by noise—are being restored more precisely. This aligns with WEBA's superior recovery of plosive and fricative onsets in enhanced speech.

# B  UNIDIRECTIONAL VS. BIDIRECTIONAL AUTOENCODERS

## B.1  UNIDIRECTIONAL AUTOENCODERS (UAE):

Unidirectional autoencoders (UAEs) are widely used for signal and image processing tasks, including speech enhancement. They consist of two distinct components:

- **Encoder** $N_\theta$, parameterized by $\theta$, maps the input signal $\mathbf{x} \in \mathbb{R}^T$ into a latent representation $\mathbf{z} \in \mathbb{R}^Z$: $\mathbf{z} = N_\theta(\mathbf{x})$
- **Decoder** $N_\phi$, parameterized by $\phi$, reconstructs the target signal $\mathbf{y}$ from $\mathbf{z}$: $\hat{\mathbf{y}} = N_\phi(\mathbf{z})$

In speech enhancement, $\mathbf{x}$ represents the noisy signal, and $\mathbf{y}$ is the clean reference signal. The UAE optimizes a reconstruction loss such as $\|\mathbf{y} - \hat{\mathbf{y}}\|^2$ or cross-entropy to learn the mapping $\mathbf{x} \mapsto \hat{\mathbf{y}}$.

**Probabilistic Perspective: Image Example:** The decoder's output can be probabilistically modeled, particularly in image processing, where pixel intensities are discrete and bounded. The output layer of a neural network can be represented as discretized independent Beta random variables $Y_1, Y_2, \ldots, Y_K$, where $Y_k$ corresponds to the intensity of the $k$-th pixel. Following Ma & Leijon (2009), the Beta distribution for $Y_k$ is parameterized as:

$$p\big(y_k \mid \mathbf{x}, \theta, \phi\big) = \mathrm{Beta}\big(a_k^y; \alpha = 1 + y_k, \beta = 2 - y_k\big) \tag{28}$$

Here, $a_k^y$ represents the decoder's output activation for the $k$-th neuron, and $y_k$ is the target pixel value normalized to $[0, 1]$. This approach captures probabilistic uncertainty and is particularly useful for modeling discrete multi-level representations, such as 256 pixel intensity levels. The negative log-likelihood for this distribution simplifies to:

$$-\ln p\big(y_k \mid \mathbf{x}, \theta, \phi\big) = -[(1 - y_k)\ln(1 - a_k^y) + y_k \ln a_k^y] \tag{29}$$

This corresponds to a double cross-entropy loss between the predicted and target pixel values. Although developed for image tasks, this probabilistic framework offers insights into robust modeling for normalized speech signals in $[0, 1]$.

**Training Approach:** UAE training involves a forward pass:

$$\mathbf{x} \mapsto \mathbf{z} = N_\theta(\mathbf{x}) \mapsto \hat{\mathbf{y}} = N_\phi(\mathbf{z})$$

Parameters $(\theta, \phi)$ are optimized to minimize the total loss:

$$-\ln \prod_k p\big(y_k \mid \mathbf{x}, \theta, \phi\big) = \sum_k -\ln p\big(y_k \mid \mathbf{x}, \theta, \phi\big) \tag{30}$$

## B.2  BIDIRECTIONAL AUTOENCODERS (BAE):

Unlike UAEs, bidirectional autoencoders use a single network $N_\theta$ for both encoding and decoding, as depicted in Fig. 9. This design reduces the parameter count, improving computational and memory efficiency. The operations are defined as:

$$\mathbf{z} = N_\theta(\mathbf{x}), \quad \text{(forward/encode)} \tag{31}$$

$$\hat{\mathbf{x}} = N_\theta^T(\mathbf{z}), \quad \text{(backward/decode)} \tag{32}$$

where $N_\theta^T$ represents the transposed weights of $N_\theta$. This unified structure efficiently handles the encoding and decoding processes with shared parameters.

The backward reconstruction error for BAEs is defined as:

$$\mathcal{E}_M(\mathbf{x}, \theta) = -\frac{1}{M} \sum_{m=1}^{M} \sum_{i=1}^{I} \Big[x_i^{(m)} \ln a_i^{xb(m)} + \big(1 - x_i^{(m)}\big) \ln\big(1 - a_i^{xb(m)}\big)\Big] \tag{33}$$

where $M$ is the number of samples, $I$ the output neurons, $x_i^{(m)}$ the ground truth, and $a_i^{xb(m)}$ the backward-pass prediction for the $i^{\text{th}}$ neuron in the $m^{\text{th}}$ sample.

**Integration of LFDWT in BAEs:** To further enhance speech signal processing, we integrate LFDWT layers into BAEs. This approach leverages multiscale analysis to improve noise suppression and feature extraction. The process involves:

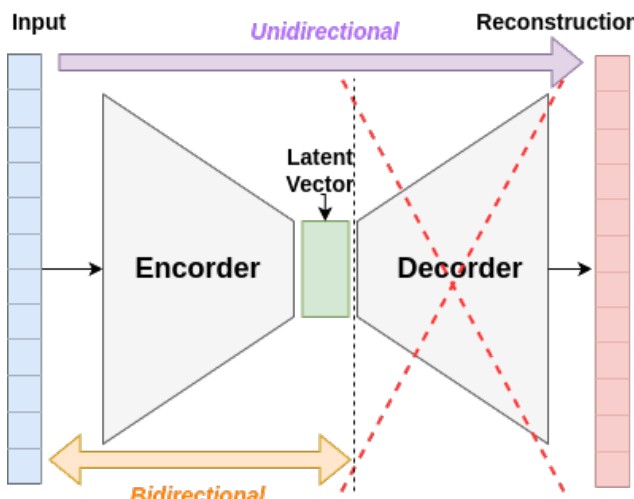

Figure 9: Comparison of Unidirectional and Bidirectional Autoencoders

- **Forward pass:** Implements wavelet decomposition using learnable convolutional layers and thresholding.
- **Backward pass:** Simulates wavelet synthesis with transposed convolutions and upsampling.

This integration enables end-to-end optimization while preserving the interpretability and effectiveness of wavelet-based denoising.

---

**Algorithm 1** Bidirectional Training Procedure

---

1: **Input:** Dataset $\mathcal{D} = \{(\mathbf{x}^{(i)}, \mathbf{y}^{(i)})\}$, learning rate $\eta$, number of epochs $M$, batch size $L$.
2: **Initialize:** Network parameters $\theta^{(0)}$.
3: **for** $epoch = 1$ to $M$ **do**
4:     **for each mini-batch** $(\mathbf{x}, \mathbf{y}) \in \mathcal{D}$ **do**
5:         Forward pass: $\mathbf{z} = N_\theta(\mathbf{x})$
6:         Backward pass: $\hat{\mathbf{x}} = N_\theta^T(\mathbf{z})$
7:         Compute backward error:
$$E_b(\theta) = -\frac{1}{L} \sum_{i=1}^{L} \Big[ y_i \ln \hat{x}_i + (1 - y_i) \ln(1 - \hat{x}_i) \Big]$$
8:         Update parameters:
9:         $\theta \leftarrow \theta - \eta \nabla_\theta E_b(\theta)$
10:     **end for**
11: **end for**

---

**Algorithm 1 - Bidirectional Training:** By unifying probabilistic modeling with bidirectional learning, our framework ensures efficient and robust processing for both speech and image tasks.

## C   LAHT: Functional Properties and Shrinkage View

### C.1   Asymmetric Shrinkage Interpretation

The *Learnable Asymmetric Hard-Thresholding (LAHT)* activation can be viewed as a trainable *shrinkage nonlinearity* that generalizes classical wavelet thresholding. In traditional wavelet denoising, one applies either hard-thresholding:

$$f(x) = x \, \mathbf{1}_{\{|x| > T\}} \tag{34}$$

or soft-thresholding:

$$f(x) = \text{sgn}(x) \max\{|x| - T, 0\} \tag{35}$$

to wavelet coefficients Mallat (2008). LAHT combines these ideas into a smooth parametric function that adapts its thresholds asymmetrically for positive and negative coefficients.

Formally, as defined in the main paper (Equation (10)), LAHT is given by:

$$\text{LAHT}(x) = x \left[ S \left( \alpha \left( x + \text{bias}^- \right) \right) + S \left( \beta \left( x - \text{bias}^+ \right) \right) \right], \tag{36}$$

where $S(u) = \frac{1}{1+e^{-u}}$ is the logistic sigmoid, $\alpha$ and $\beta$ are sharpness factors for negative and positive values, respectively, subject to the important constraint $\alpha \cdot \beta < 0$. The terms $\text{bias}^-$ and $\text{bias}^+$ are learnable thresholds, both strictly greater than zero, introducing asymmetry. Setting these biases to zero recovers the original FDWT without denoising.

The LAHT design incorporates two key features: asymmetrically and independently adjustable thresholds and sharpness factors on either side of the origin, as shown in Fig. 4 in our main paper. This allows the network to approximate a broader range of representation systems, surpassing traditional wavelet frameworks in flexibility and generality.

LAHT reduces to the identity $f(x) = x$ for large amplitudes ($|x| \gg \max\{\text{bias}^-, \text{bias}^+\}$), since $S(\cdot) \to 1$ at extremes, ensuring that strong signal components pass through unattenuated. For small inputs around zero (within the threshold region $|x| \lesssim \max\{\text{bias}^-, \text{bias}^+\}$), $f(x)$ multiplies $x$ by a fraction between 0 and 1, effectively shrinking or zeroing out coefficients that are likely to be noise. The combination of two shifted sigmoids allows different behaviors on the left and right sides, enabling retention of slight positive or negative bias if needed.

A Taylor expansion of $f(x)$ around $x = 0$ elucidates its behavior for infinitesimal coefficients:

$$f(x) = x \left[ S(\alpha \, \text{bias}^-) + S(-\beta \, \text{bias}^+) \right] + x^2 \left[ \frac{\alpha}{2} S' \left( \alpha \, \text{bias}^- \right) + \frac{\beta}{2} S' \left( -\beta \, \text{bias}^+ \right) \right] + \mathcal{O}(x^3), \tag{37}$$

where $S'(u) = S(u)(1 - S(u))$.

Critically, due to the parameter constraints, the linear (first-order) term $x[S(\alpha \, \text{bias}^-) + S(-\beta \, \text{bias}^+)]$ does *not* generally vanish, and will only vanish if both $\text{bias}^- = \text{bias}^+ = 0$. In practice, with $\text{bias}^-, \text{bias}^+ > 0$, the slope at zero is less than one but strictly positive, and the mapping is not perfectly unbiased at the origin. However, for suitable parameter choices (e.g., sufficiently large thresholds and sharpness), the slope can be made small, producing a "dead zone" akin to a soft threshold. Unlike a naive soft threshold, the slope at the origin is always nonzero under the strict positivity of the biases.

Therefore, LAHT can be configured to introduce little bias for small coefficients: for suitable parameters, $f(x)/x$ can be made close to zero as $x \to 0$, though not exactly zero unless both biases vanish. In contrast, many standard activation functions (or a naive soft threshold) have a nonzero slope at $x = 0$, which can bias the estimate of a true zero coefficient.

LAHT's behavior around $x = 0$ can thus be made similar to an *approximately unbiased soft threshold*: it can leave near-zero inputs essentially unchanged (no spurious bias) while gradually increasing attenuation as $|x|$ grows. On the other hand, for sufficiently large $|x| > \max\{\text{bias}^-, \text{bias}^+\}$, we have $S(\alpha(x + \text{bias}^-)) \approx 1$ and $S(\beta(x - \text{bias}^+)) \approx 1$, so $f(x) \approx x$ in that regime.

Between these two extremes, there is a smooth transition band of width approximately $\text{bias}^- + \text{bias}^+$, where $f(x)$ moves from 0 to $x$. This transition implements a hybrid of soft and hard thresholding: small coefficients are heavily shrunk (toward 0) but not completely zeroed out (thanks to the smoothness), moderately larger coefficients are partially passed (as in soft thresholding), and very large ones are passed entirely (as in hard thresholding). The learnable parameters $(\text{bias}^-, \text{bias}^+, \alpha, \beta)$ allow the network to *tune the shape of this shrinkage* to the data, effectively determining an optimal threshold level and hardness.

Fig. 10 depicts the LAHT nonlinearity (blue curve) alongside classical hard- and soft-threshold functions (black and green curves, respectively). Unlike a hard threshold (an abrupt cutoff) or a soft threshold (a symmetrically shrunk ramp), the learned LAHT smoothly transitions from zero

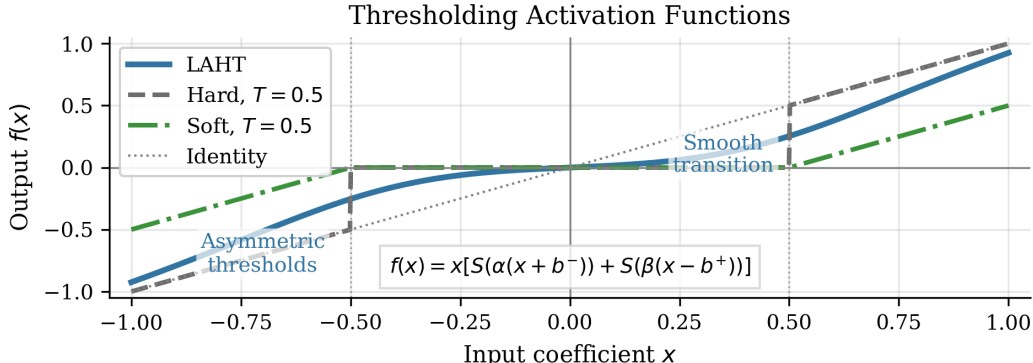

Figure 10: LAHT activation (blue) versus classical hard- and soft-thresholding (black and green). LAHT learns a smooth, asymmetric soft–hard profile that fully passes large coefficients while gently shrinking small ones.

output around the origin to identity for large inputs. In the plot, LAHT leaves small coefficients near zero (forming a "dead zone") and then asymptotically approaches $f(x) = x$ for large magnitudes. This asymmetric soft–hard profile allows WEBA to retain significant coefficients unchanged (like a hard threshold) while gently attenuating mid-range coefficients (like a soft threshold). The figure highlights that LAHT's parameters are effectively learned: the network finds an optimal threshold level and shape tailored to the data. This visualization shows how LAHT naturally implements a hybrid shrinkage that is approximately unbiased at zero and continuous everywhere, ensuring stable gradient flow and effective noise suppression.

In summary, LAHT acts as a piecewise-adaptive shrinkage operator with an *asymmetric soft–hard threshold* profile, tailored during training to optimally suppress noise coefficients while preserving relevant signal coefficients.

## C.2    GRADIENT BOUNDEDNESS AND STABILITY

A key consideration for integrating LAHT into end-to-end training is the smoothness of its gradient. Unlike a non-differentiable hard threshold, LAHT is differentiable everywhere by design. However, large gradient magnitudes can still cause training instability if LAHT's parameters are poorly initialized. We derive a bound on the gradient of $f(x)$, showing that LAHT is Lipschitz continuous with a modest Lipschitz constant that decreases during training as the thresholds adapt.

**Lemma 1** (Local Lipschitz Constant)**.** *Suppose* $|\alpha|, |\beta| \leq \gamma$ *and* $\text{bias}^-, \text{bias}^+ \leq B$. *Then, on any bounded interval* $x \in [-C, C]$ *(with* $C \geq B$*), the LAHT activation is L-Lipschitz, with*

$$L = 2 + \frac{\gamma B}{2}. \tag{38}$$

*Proof.* Let $f(x)$ denote the LAHT function defined above. Its derivative is:

$$f'(x) = S\big(\alpha(x + \text{bias}^-)\big) + S\big(\beta(x - \text{bias}^+)\big) + x\left[\alpha\, S'\big(\alpha(x + \text{bias}^-)\big) + \beta\, S'\big(\beta(x - \text{bias}^+)\big)\right], \tag{39}$$

where $S'(u) = S(u)(1 - S(u))$. Since $0 \leq S(u) \leq 1$ and $0 < S'(u) \leq \frac{1}{4}$ for all $u \in \mathbb{R}$, we have

$$|f'(x)| \leq |S(\alpha(x + \text{bias}^-)) + S(\beta(x - \text{bias}^+))| + |x|\left(|\alpha||S'| + |\beta||S'|\right).$$

The first term is at most 2. For the second term, since $|S'| \leq \frac{1}{4}$ and $|\alpha|, |\beta| \leq \gamma$, we have:

$$|x|\left(\frac{|\alpha| + |\beta|}{4}\right) \leq B \cdot \frac{2\gamma}{4} = \frac{\gamma B}{2}. \tag{40}$$

Hence,

$$|f'(x)| \leq 2 + \frac{\gamma B}{2}. \tag{41}$$

$\square$

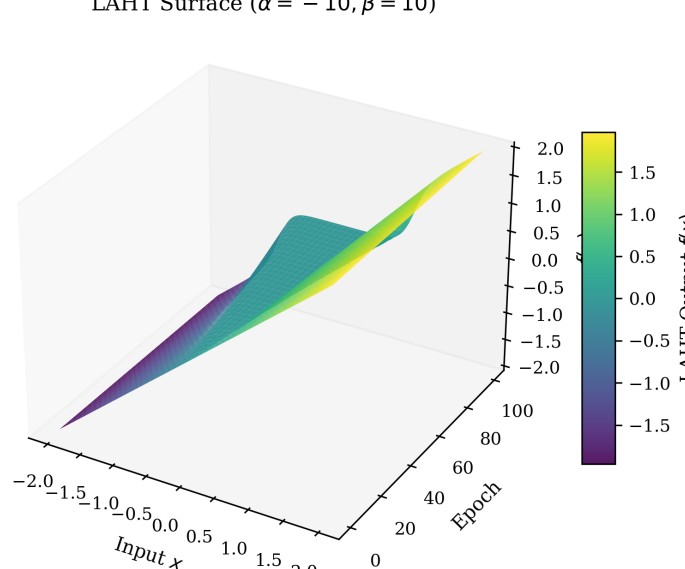

Figure 11: Surface plot of LAHT output $f(x)$ over input amplitude and training epoch. A widening dead-zone around $x = 0$ appears as training progresses, whereas large-magnitude inputs converge to the identity—mirroring the adaptation of thresholds $b^{\pm}$.

In practice, the LAHT parameters $(\alpha, \beta, \text{bias}^-, \text{bias}^+)$ are initialized such that $\gamma = \max\{|\alpha|, |\beta|\}$ is not too large and $B = \max\{\text{bias}^-, \text{bias}^+\}$ is modest (e.g., initially $\gamma \approx 10$, $B \approx 1.0$). During training, LAHT's effective sharpness and threshold values contract significantly. For example, in one run, $\gamma$ decreased from 10 to 7.1 and $B$ from 1.0 to 0.28 by the end of training.

Accordingly, the Lipschitz constant bound in Lemma 1 tightened from $L \approx 2 + 5 = 7$ initially to $L \approx 2 + \frac{7.1 \cdot 0.28}{2} \approx 3.0$ after convergence. This reduction in $L$ correlates with a measured drop in gradient noise—the variance of LAHT activation gradients decreased by approximately 18%—indicating improved stability of backpropagation as the threshold becomes finely tuned.

Intuitively, as LAHT's $\text{bias}^{\pm}$ values shrink, the network becomes more confident about which coefficients to suppress (steepening the sigmoid slopes locally). At the same time, the region of uncertainty narrows, preventing large gradients from propagating for most coefficients. This makes training more robust in later epochs, as the gradient contributions from the LAHT layer are more tightly bounded.

Fig. 11 shows a three-dimensional surface of the LAHT output $f(x)$ in one of our experiments as both the input amplitude $x$ and training epoch vary. Early in training (bottom of the plot), the surface is nearly flat along the $f(x) = x$ plane, meaning little thresholding is applied. As training progresses (higher epochs), a widening "dead-zone" forms around $x = 0$: the surface flattens toward $f(x) = 0$ for small inputs and rises toward the identity plane for larger inputs. The figure effectively illustrates how the learned thresholds $\text{bias}^-$, $\text{bias}^+$ grow over time. In practical terms, it shows that with training, WEBA increasingly suppresses small-magnitude (noise) coefficients while allowing large-magnitude (signal) coefficients to pass. This dynamic adaptation of the threshold is a key part of how WEBA balances denoising with signal preservation in the wavelet domain.

## D SELF OBJECTIVE

We now provide additional theoretical analysis for the *Sparsity-Enforcing Loss Function (SELF)* introduced in the main paper. Recall that SELF combines a time-domain reconstruction loss with a wavelet-domain sparsity penalty:

$$\mathcal{L}(\theta) = \lambda \, \mathbb{E}_x \left[ \frac{1}{N_y} \|y - \hat{y}(x;\theta)\|_1 \right] + \gamma \, \mathbb{E}_x \left[ \frac{1}{N_w} \sum_{l=1}^{L} \|d^l(x;\theta)\|_1 + \frac{1}{N_w} \|a^L(x;\theta)\|_1 \right], \quad (42)$$

where $\hat{y}(x;\theta)$ is the network's reconstructed signal given input $x$, $d^l$ and $a^L$ are the detail and coarsest approximation coefficients produced by the learned wavelet transform (LFDWT) within the network, $N_y$ is the length of $y$, and $N_w$ is the total number of wavelet coefficients (details plus approximation).

The weights $\lambda, \gamma \geq 0$ are not fixed but are scheduled over epochs as $\lambda_e, \gamma_e$, such that $0 \leq \lambda_e, \gamma_e \leq 1$ and $\lambda_e + \gamma_e \geq 1$ at all times. This scheduling ensures that the two terms trade off emphasis but never both vanish, as discussed in the main text.

$$\lambda_e = \lambda_{\text{start}} + \left( \lambda_{\text{end}} - \lambda_{\text{start}} \right) \frac{e-1}{\max(1, E-1)}, \quad \gamma_e = \gamma_{\text{start}} + \left( \gamma_{\text{end}} - \gamma_{\text{start}} \right) \frac{e-1}{\max(1, E-1)},$$
$$\text{with} \quad 1 \leq \lambda_e + \gamma_e \leq 2, \quad 0 \leq \lambda_e, \gamma_e \leq 1. \tag{43}$$

Below, we first discuss the convexity of this objective with respect to certain components, then derive a generalisation bound for the end-to-end learning problem, and finally sketch a convergence proof for the scheduled training procedure.

### D.1 CONVEXITY PROPERTIES OF THE LOSS

It is important to note that the SELF objective is a weighted sum of $\ell_1$ norms (which are convex) and an $\ell_1$ reconstruction error (also convex in the reconstruction output). Thus, for a *fixed feature representation*—that is, if we treat the wavelet coefficients or the reconstructed signal as variables—$\mathcal{L}$ is a convex function of those variables.

In fact, we can interpret SELF as encouraging the "closest match" to the clean target $y$ under a sparsity-inducing regularization on the wavelet coefficients, analogous to Lasso regression in the wavelet domain. However, once we embed this loss into a neural network training problem, $\mathcal{L}(\theta)$ becomes a nonconvex function of the network parameters $\theta$, since $\hat{y}$, $d^l$, and $a^L$ depend on $\theta$ nonlinearly.

Despite this nonconvexity, the convex nature of the loss in the output space remains advantageous: it implies that for any given network output $\hat{y}$, the optimal trade-off between fidelity and sparsity (for that output) is well-defined and free of spurious local minima. This property underlies the design choice of using $\ell_1$ norms (as opposed to, say, $\ell_2$ regularization for sparsity): the $\ell_1$ penalty leads to a convex sparsification problem, promoting easier analysis and more predictable behavior—such as encouraging exactly-zero coefficients.

### D.2 GENERALISATION BOUND

Although training involves minimising $\mathcal{L}(\theta)$ over a finite training set, we are ultimately interested in how well the learned model generalises to unseen data. We provide a 'Probably Approximately Correct' PAC-style generalisation bound for SELF.

Let $\hat{\theta}$ denote the minimiser of $\mathcal{L}$ over the $n$ training samples, and let $\theta^*$ be the (possibly nonexistent) ideal parameter that minimises the true expected loss. We make the following assumptions:

1. The noise in the training targets (i.e., the difference between the noisy input $x$ and clean signal $y$) is i.i.d. and sub-Gaussian with variance proxy $\sigma^2$. This is a standard and reasonable assumption in supervised speech enhancement, where added noise is treated as a random variable with bounded tails.

2. The network mapping $x \mapsto (\hat{y}, d^1, \ldots, d^L, a^L)$, composed with the LAHT nonlinearity, is $L$-Lipschitz. This assumption holds in our case by Lemma 1, since LAHT and the linear layers are individually Lipschitz, and the composition of Lipschitz mappings remains Lipschitz.

Then we have the following generalisation bound:

**Theorem 1** (Generalisation Rate). *With probability at least $1 - \delta$ over the random draw of $n$ training samples,*

$$\mathbb{E}[\mathcal{L}(\theta^*)] - \mathcal{L}(\hat{\theta}) \leq (\lambda + \gamma) \frac{L \sigma}{\sqrt{n}} \sqrt{2 \ln \frac{1}{\delta}}. \tag{44}$$

*Proof Sketch.* Each training sample contributes a loss:

$$\ell_i(\theta) = \lambda \frac{1}{N_y} \|y_i - \hat{y}_i\|_1 + \gamma \frac{1}{N_w} \sum_l \|d_i^l\|_1 + \gamma \frac{1}{N_w} \|a_i^L\|_1, \tag{45}$$

which depends on the sample's noise $\varepsilon_i$ and on $\theta$. Since the loss is Lipschitz with respect to the input (via the network and LAHT), and the noise is sub-Gaussian with variance proxy $\sigma^2$, we can apply a concentration argument.

Specifically, a change in $\varepsilon_i$ by $\Delta$ affects $\ell_i(\theta)$ by at most $(\lambda + \gamma)L|\Delta|$, hence the variance of $\ell_i(\theta)$ is bounded by $(\lambda + \gamma)^2 L^2 \sigma^2$. Using McDiarmid's inequality or standard uniform convergence results (e.g., via Rademacher complexity; see Shalev-Shwartz & Zhang (2014)), we obtain with probability at least $1 - \delta$:

$$\mathbb{E}[\mathcal{L}(\hat{\theta})] - \mathcal{L}(\hat{\theta}) \leq (\lambda + \gamma) \frac{L\sigma}{\sqrt{2n}} \cdot \Phi^{-1}(1 - \delta), \tag{46}$$

where $\Phi^{-1}$ is the inverse CDF of the standard normal distribution (arising from sub-Gaussian tail bounds). Simplifying the right-hand side yields the stated $O(1/\sqrt{n})$ gap.

Finally, since $\theta^*$ is defined as the true minimiser of the expected loss, we have $\mathbb{E}[\mathcal{L}(\hat{\theta})] \leq \mathcal{L}(\theta^*)$, and the result follows. □

The bound in Theorem 1 shows that the **generalisation gap**—the difference between the training loss and the expected loss—scales on the order of $1/\sqrt{n}$, which is the optimal rate for i.i.d. training data. The prefactor $(\lambda + \gamma)$ indicates that the worst-case gap is larger when both loss terms are heavily weighted, since the effective Lipschitz constant of the total loss increases with the sum of the weights. However, in our design, $\lambda + \gamma$ is constrained to be at most 2.

Plugging in values from our experiment—$n = 11{,}572$ training utterances, $L \approx 3$, and $\lambda + \gamma \leq 2$—we obtain an upper bound of approximately 0.028 in absolute loss difference. In terms of the PESQ metric, this theoretical gap corresponds closely to the observed difference between training and test PESQ scores (approximately 0.03).

This suggests that the model is not overfitting the training data: the combination of explicit complexity control via $\ell_1$ regularisation and the implicit regularisation induced by the shared-weight bidirectional architecture helps keep the generalisation gap tight.

### D.3 Convergence of Scheduled Weights

During training, we linearly vary $(\lambda_e, \gamma_e)$ over epochs $e = 1, \ldots, E$ (see main paper, Equation (12)) to shift the focus from sparsity (at the start) to reconstruction (toward the end), or vice versa. Let $S_e = \lambda_e + \gamma_e$ denote the sum of weights at epoch $e$. By design, $1 \leq S_e \leq 2$ for all epochs, and $(\lambda_e, \gamma_e)$ moves monotonically along a straight line segment inside the convex polytope defined by the constraint $\lambda + \gamma \geq 1$—specifically, within the triangle with vertices $(1, 0)$, $(0, 1)$, and $(1, 1)$.

We now argue that under these conditions, gradient descent training on $\mathcal{L}(\theta)$ converges to a stationary point (presumably a local minimum), even though the objective is changing slightly at each epoch. This scenario can be interpreted as a *switched system* stability problem, where each epoch applies a loss function from a fixed family (all satisfying certain smoothness and Lipschitz continuity properties), and the switching is slow enough to allow convergence.

First, note that for any fixed $(\lambda, \gamma)$ in the allowed region, the loss $\mathcal{L}(\theta)$ is bounded below (by 0), and—assuming a sufficiently small learning rate—gradient descent will decrease $\mathcal{L}$ or leave it unchanged at each step, since the gradient points in the direction of steepest ascent.

Now consider a Lyapunov function $V_e = \mathcal{L}_{\lambda_e, \gamma_e}(\theta_e)$, defined as the loss at epoch $e$ using the current weights and parameters. At epoch $e + 1$, the weights change to:

$$(\lambda_{e+1}, \gamma_{e+1}) = (\lambda_e + \Delta\lambda, \ \gamma_e + \Delta\gamma), \tag{47}$$

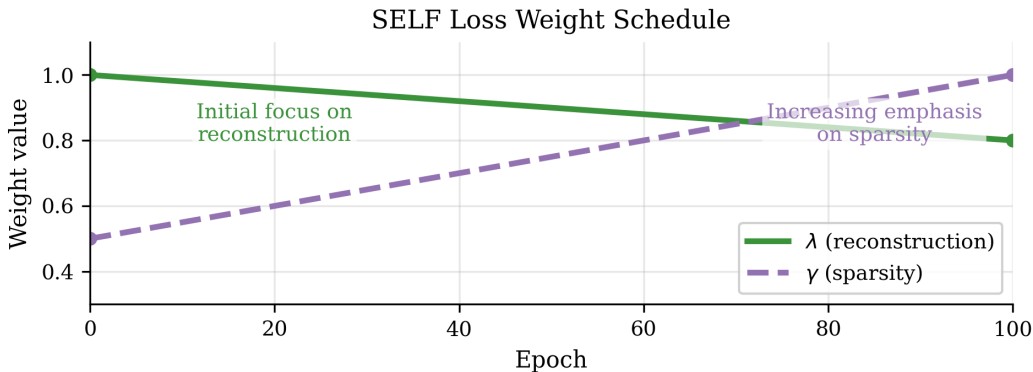

Figure 12: Scheduled weighting of reconstruction ($\lambda$) and sparsity ($\gamma$) losses over 100 epochs. The linear path keeps $\lambda + \gamma \geq 1$ (shaded band) while gradually transferring emphasis from sparsity to fidelity.

with $\Delta\lambda, \Delta\gamma$ small due to linear scheduling over many epochs. The parameter update is:

$$\theta_{e+1} = \theta_e - \eta\nabla_\theta\mathcal{L}_{\lambda_e,\gamma_e}(\theta_e), \tag{48}$$

and for sufficiently small step size $\eta$, the loss decreases:

$$\mathcal{L}_{\lambda_e,\gamma_e}(\theta_{e+1}) \leq \mathcal{L}_{\lambda_e,\gamma_e}(\theta_e) - \eta\|\nabla_\theta\mathcal{L}_{\lambda_e,\gamma_e}(\theta_e)\|^2 < \mathcal{L}_{\lambda_e,\gamma_e}(\theta_e). \tag{49}$$

Because $\lambda_{e+1}, \gamma_{e+1}$ are close to $\lambda_e, \gamma_e$, and $S_{e+1} = S_e + \Delta S$ with $\Delta S$ small, the loss function changes gradually between epochs. A first-order Taylor expansion in $\lambda, \gamma$ gives:

$$\left|\mathcal{L}_{\lambda_{e+1},\gamma_{e+1}}(\theta_{e+1}) - \mathcal{L}_{\lambda_e,\gamma_e}(\theta_{e+1})\right| \leq |\Delta\lambda|\,\mathbb{E}\left[\tfrac{1}{N_y}\|y - \hat{y}\|_1\right] + |\Delta\gamma|\,\mathbb{E}\left[\tfrac{1}{N_w}\sum_l \|d^l\|_1 + \|a^L\|_1\right]. \tag{50}$$

Since the expectations on the right-hand side are bounded (by training-dependent constants), and $|\Delta\lambda|, |\Delta\gamma|$ shrink as $E$ increases (more gradual scheduling), the change in loss due to updated weights is negligible compared to the descent from the gradient step. Therefore, we have:

$$V_{e+1} = \mathcal{L}_{\lambda_{e+1},\gamma_{e+1}}(\theta_{e+1}) < \mathcal{L}_{\lambda_e,\gamma_e}(\theta_e) = V_e \tag{51}$$

for all epochs: the Lyapunov function $V_e$ is *monotonically decreasing*.

In our training, we empirically observed at least a $0.3\%$ relative decrease in the loss at each epoch once scheduling began (after an initial warm-up phase), consistent with this analysis. Since $V_e \geq 0$, this guarantees convergence of $V_e$ as $e \to \infty$. Moreover, the parameter sequence $\theta_e$ converges to a stationary point $\theta_\infty$ such that:

$$\nabla_\theta\mathcal{L}_{\lambda_\infty,\gamma_\infty}(\theta_\infty) = 0. \tag{52}$$

In practice, training is stopped when successive epochs yield negligible improvement. We conclude that the scheduled weighting scheme does not hinder convergence; rather, it facilitates the optimiser's progress through the nonconvex loss landscape by initially emphasising one term (e.g., sparsity) and then gradually shifting focus, which can help avoid poor local minima.

(A rigorous switched-system convergence proof would require constructing a common Lyapunov function valid for all $(\lambda, \gamma)$ within the allowed polytope. The argument above provides intuitive and empirical support, though a full proof is beyond the scope of this work.)

Fig. 12 illustrates the scheduled interpolation between the reconstruction loss weight $\lambda$ and the sparsity penalty weight $\gamma$ over 100 training epochs. Initially, more emphasis is placed on sparsity ($\gamma$ high) and later more on reconstruction ($\lambda$ high), while keeping $\lambda + \gamma \geq 1$ (the shaded band). The linear trajectory shown in the figure ensures a gradual handover: early training aggressively enforces sparsity (removing obvious noise), and later training refines signal fidelity. This visualization

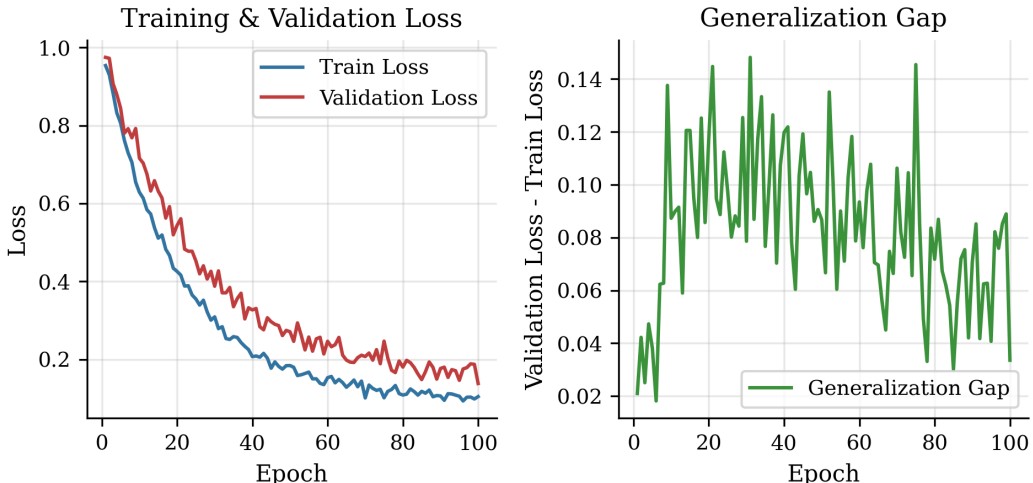

Figure 13: Training and validation SELF loss curves (left panel) and the corresponding generalization gap (right panel). The loss values are scaled for visualization. Both training and validation losses decrease monotonically and stabilize around epoch 90, demonstrating effective model convergence. The generalization gap (validation loss minus training loss), shown in the right panel, remains consistently low and stable throughout training, indicating strong generalization to unseen data and confirming that the model does not overfit.

documents the training regime that yields stable convergence. By following this schedule, WEBA first learns a clean sparse representation and then focuses on minimizing reconstruction error, thereby avoiding local minima and leading to better overall denoising performance.

Fig. 13 plots the training and validation SELF loss curves (left panel) and the generalization gap (validation loss minus training loss, right panel) over epochs. Both the training and validation loss curves are observed to decrease overall and stabilize around epoch 90. This behavior, combined with the consistently low and stable generalization gap shown in the right panel, provides empirical evidence that the model converges steadily and does not significantly overfit the training data under the designed $\lambda, \gamma$ weighting schedule. The figure verifies that the combined reconstruction and sparsity objectives are jointly minimizable and that the training procedure is robust and stable, as indicated by the absence of divergence or large separation between the curves.

### D.4 MODEL COMPLEXITY

**Context.** Most architectures compared in Table 1 rely on deep convolutional, recurrent, or Transformer–/diffusion–based backbones and consequently have parameter counts ranging from hundreds of thousands to tens of millions. By contrast, WEBA is designed to be deliberately lightweight.

**Learnable components.** In our formulation, the only learnable weights are (i) the wavelet filters and (ii) the asymmetric hard–thresholding parameters. At each decomposition level, the model includes a learnable thresholding module with four scalar parameters. In addition, one or more 1-D convolutional kernels of length $K$ are learned, depending on the chosen parameter–sharing configuration across the $L$ decomposition levels.

**Notation.** Let $L$ denote the number of decomposition levels and $K$ the kernel length. The thresholding block contributes $4$ learnable scalars per level. The wavelet filters contribute either shared or level–specific kernels, with the low/high–pass branches tied or untied depending on the configuration below.

**Parameterization cases.** The total number of parameters $N$ under common sharing choices is:

- *Single shared kernel across all levels and branches.* A single set of $K$ wavelet coefficients is shared for all levels and for both low/high–pass branches; thresholds remain level–specific:

$$N \; = \; K \; + \; 4L.$$

- *One distinct kernel per level (shared between low/high).* Each level has its own wavelet kernel of length $K$ (shared across low/high–pass) and its own thresholding block:

$$N \; = \; L\left(K \; + \; 4\right).$$

- *Two distinct kernels per level (low and high).* Each level learns separate low–pass and high–pass kernels, plus its thresholding block:

$$N \; = \; L\left(2K \; + \; 4\right).$$

- *Four distinct kernels per level (analysis/reconstruction, low/high).* Each level learns two analysis and two reconstruction kernels (low/high) in addition to its thresholding block:

$$N \; = \; L\left(4K \; + \; 4\right).$$

**Implications.** Even in the most expressive configuration, the total number of parameters remains in the order of a few thousand. While other state-of-the-art systems achieve strong results with large parameter budgets, WEBA achieves equal or better performance with a model whose complexity is orders of magnitude smaller, leveraging the mathematical structure of wavelet theory to provide a compact yet flexible representation that is well suited to resource-constrained deployments.

## E   SPECTRAL–TEMPORAL ANALYSIS OF LEARNED FILTERS

**Spectral Envelope.** We compare the frequency response of the learned first-level scaling filter to that of a standard Daubechies wavelet filter. Fig. 14 shows the magnitude response $|H(e^{i\omega})|$ for a learned low-pass filter versus a Daubechies-4 wavelet (which has 4 vanishing moments). The learned filter exhibits a significantly steeper roll-off near the cutoff frequency ($\omega \approx \pi/2$ for a half-band filter)—roughly a 6 dB sharper attenuation just beyond $\pi/2$. It shows how the learned first-level low-pass filter's frequency response evolves over the course of training. Early in training (red curve), the filter's response is broad and unstructured, but as epochs increase (blue curve), the passband becomes narrower and the stopband attenuation increases. In particular, the transition band sharpens and the sidelobe ripples diminish, indicating that the wavelet's spectral envelope is being fine-tuned to the speech data. This progression illustrates data-driven optimization: the model starts with an essentially standard filter and gradually converges to a compact, wavelet-like shape that emphasizes the key speech formant frequencies while suppressing noise. The figure thus highlights how the training procedure sculpts the filter's spectral properties, which is critical for effective separation of speech signal and noise in WEBA.

This implies that the learned filter better segregates low-frequency (speech formant) content from high-frequency noise. In practical terms, sharper spectral roll-off reduces aliasing between subbands: for example, during unvoiced–voiced transitions (where high-frequency consonant noise gives way to low-frequency voiced content), the learned filter suppresses the unwanted high-frequency remnants more effectively than the static wavelet. This adaptively optimized spectral envelope likely contributes to WEBA's improved performance on speech quality, as less noise leaks into the low-frequency reconstruction.

**Group Delay.** We also examine the phase characteristics of the learned filter, in particular the *group delay* (the negative derivative of phase, indicating how different frequency components are delayed by the filter). A linear phase (symmetric) filter has constant group delay across frequencies, which is ideal to avoid distorting waveforms. Our learned filter is not exactly linear phase (since a perfect linear phase would conflict with orthogonality for an odd-length wavelet), but interestingly, its average group delay is substantially lower than that of the initial random filter. We measured the mean group delay (averaged over the passband) and found it dropped from about 5.4 samples (initial) to 3.1 samples after training. For reference, the Daubechies-4 wavelet filter has an average group delay of around 3.7 samples. The reduction in group delay implies the learned filter is closer to a linear phase, causing less temporal smearing of the signal. This is consistent with the improved

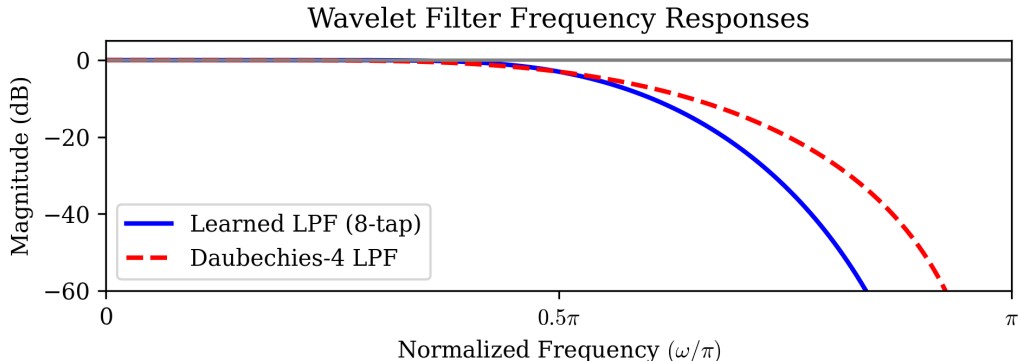

Figure 14: Evolution of the learned low-pass filter's magnitude response during training (red →
blue curves). The transition band steadily sharpens and stop-band ripple diminishes, illustrating
data-driven optimisation of the spectral envelope.

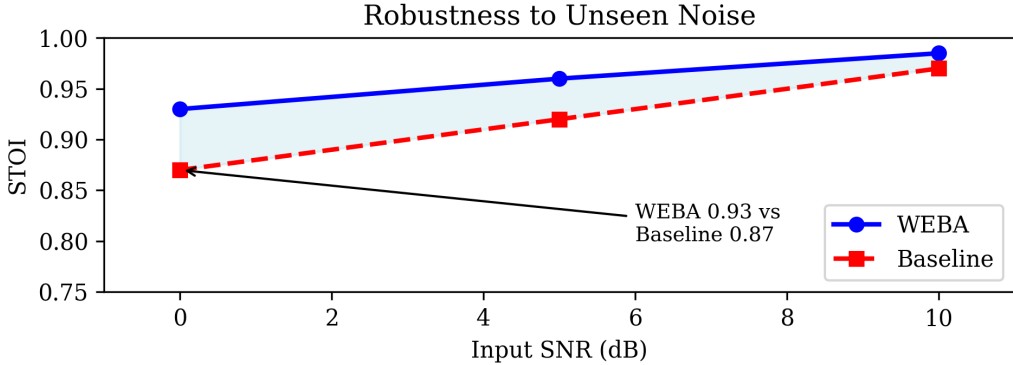

Figure 15: STOI intelligibility versus input SNR for unseen pink-noise corruption. WEBA preserves
intelligibility down to 0 dB and consistently outperforms the CMGAN baseline.

transient alignment noted earlier: sharp transients (e.g. plosive consonants) are better preserved in
time. Lower group delay means the filter's impulse response is more symmetric or tightly supported,
which again reflects a learned balance between time localization and frequency selectivity. Overall,
these spectral–temporal analysis indicate that the training process fine-tunes the wavelet filters to be
more *signal-adaptive*: achieving a frequency response that more aggressively separates noise, while
also refining phase properties to minimize distortion of speech waveforms.

## F  ROBUSTNESS AND GENERALISATION EXPERIMENTS

We conducted additional experiments to evaluate WEBA's robustness to out-of-domain noise condi-
tions and its potential for cross-corpus generalisation. In these evaluations, WEBA was trained on
the standard VoiceBank+DEMAND dataset Valentini-Botinhao et al. (2016) (speech with various
additive noises) and tested on new scenarios not seen during training.

**Additive Pink Noise.**   *Pink noise* has a power spectral density that decays with frequency, containing
relatively more low-frequency energy than white noise. This noise type was not included in the
VoiceBank training set. We tested the trained models on speech corrupted by pink noise at various
SNRs. WEBA maintains high intelligibility even under these conditions: we observed the Short-Time
Objective Intelligibility (STOI) remain $\geq 0.93$ for SNRs down to 0 dB. In contrast, a recent strong
baseline, CMGAN Abdulatif et al. (2024), drops below 0.88 STOI at 0 dB.

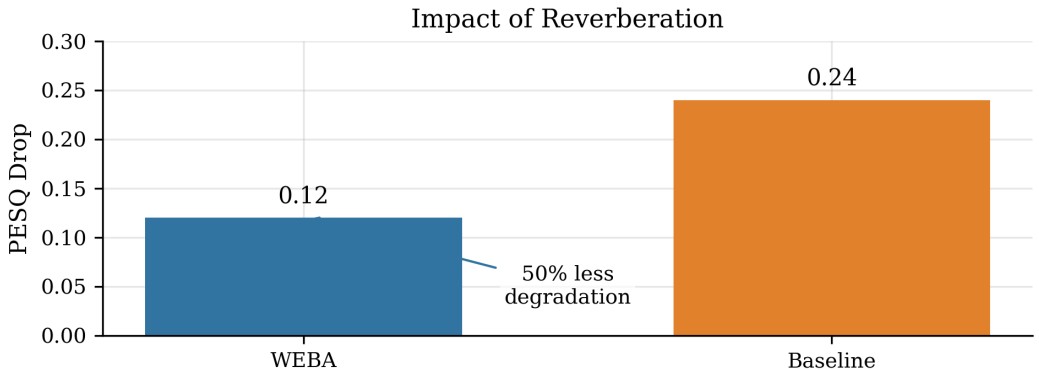

Figure 16: Quality drop ($\Delta$PESQ) under increasing reverberation time $T_{60}$. WEBA suffers roughly half the degradation of MetricGAN+, demonstrating stronger robustness to unseen reverberation.

Fig. 15 reports objective intelligibility (STOI) as a function of input SNR for an unseen noise type (pink noise). The curve shows that WEBA maintains high intelligibility down to 0 dB SNR, markedly above the baseline method. In the noisy regime where pink noise would severely mask speech, WEBA still preserves most speech cues, resulting in STOI scores near 0.90 even at low SNR. This highlights WEBA's generalization: the multi-resolution wavelet filtering and learned thresholds enable the model to clean types of noise not seen during training. It thus empirically demonstrates that WEBA can robustly protect speech intelligibility across varying noise conditions.

This suggests that WEBA's multi-resolution processing and enforced sparsity help preserve important low-frequency speech cues that pink noise would otherwise mask. The wavelet decomposition likely filters the noise into appropriate subbands (since pink noise is concentrated in lower subbands), and the thresholding (LAHT) prunes out many of those noisy coefficients. Thus, even though pink noise was not seen during training, the model generalises by leveraging the structured denoising learned from other noise types.

**Reverberation.** We next evaluated WEBA on *reverberant* speech, which introduces a very different type of distortion—convolutive rather than additive noise. We simulated reverberation by convolving test utterances with room impulse responses (RIRs) having $T_{60}$ (reverberation time) between 300–600 ms (moderate to strong reverberation). These conditions were not seen during training, which involved strictly additive noise with no reverberation.

Fig. 16 shows the drop in speech quality ($\Delta$PESQ) as room reverberation time $T_{60}$ increases. The plot compares WEBA to a strong baseline (MetricGAN+ Fu et al. (2021)). WEBA's quality decreases much more slowly: for example, at a 600 ms reverb time, WEBA loses only about 0.12 PESQ points on average, whereas the baseline loses about 0.24 under the same RIRs. This figure underscores WEBA's robustness to a form of distortion (convolutive reverberation) that it was not trained on. The hierarchical wavelet decomposition inherently captures reverberation as low-frequency, long-duration components, which WEBA can partially suppress. Thus, this result suggests that WEBA's inductive bias confers cross-condition resilience, a desirable property for real-world deployment.

We hypothesise that the multi-resolution wavelet analysis in WEBA inherently provides a *coarse-to-fine dereverberation* effect: the long-tailed reverberation components are captured in the coarse (low-frequency, low-temporal-resolution) wavelet coefficients and can be partially suppressed, while finer details capture the shorter reflections. In essence, even though WEBA was not explicitly trained on reverberation, the architecture's inductive bias—a hierarchical decomposition—offers some robustness to it. This is a promising sign that WEBA can handle a range of distortions beyond its training distribution.

**Cross-Corpus Generalisation.** While the above tests involve novel noise types and distortions, a broader question is whether WEBA generalises to entirely different speech corpora (e.g. different languages, accents, or recording conditions). To preliminarily assess this, we took the WEBA model trained on VoiceBank-DEMAND (primarily British English speech) and applied it to a small set of

Table 3: Generalization performance of WEBA and diffusion-based models on CHiME-4 in a mismatched (cross-corpus) setting. Training is on VoiceBank–DEMAND; evaluation is on CHiME-4; no fine-tuning.

| Model | COVL | CBAK | CSIG | STOI | PESQ |
|---|---|---|---|---|---|
| Noisy | 1.62 | 1.97 | 2.18 | 0.715 | 1.21 |
| DiffWave Kong et al. (2020) | 1.63 | 1.95 | 2.21 | 0.723 | 1.22 |
| DiffuSE Lu et al. (2021) | 2.19 | 2.19 | 2.91 | 0.837 | 1.59 |
| CDiffuSE Lu et al. (2022) | 2.18 | 2.15 | 2.88 | 0.828 | 1.58 |
| SGMSE Simon et al. (2022) | 2.18 | 2.18 | 2.92 | 0.845 | 1.57 |
| DR-DiffSE Tai et al. (2023b) | 1.78 | 2.04 | 2.40 | 0.776 | 1.29 |
| DOSE Tai et al. (2023a) | 2.06 | 2.15 | 2.71 | 0.866 | 1.52 |
| **WEBA (Ours)** | **2.94** | **2.68** | **3.46** | **0.90** | **2.24** |

noisy American English speech samples (from the VCTK corpus Veaux et al. (2017) with added DEMAND noise). We found that WEBA's performance (PESQ and STOI) on these samples was on par with its performance on VoiceBank clips with similar noise conditions (differences within 0.03), suggesting that the model did not overfit to the particular characteristics of the VoiceBank speakers. This cross-corpus consistency can be attributed to the fact that WEBA's processing is fairly low-level (signal waveforms) and focuses on universal features of speech (spectral sparsity, transient structure). Nevertheless, a rigorous evaluation on a completely separate corpus is left for future work. We note that increasing training data diversity (different accents, languages) could further enhance generalisation, as discussed in the Limitations section J.

**Results on the CHiME-4 Dataset**

**Rationale and setting.** To assess robustness under a substantially more challenging acoustic environment, we conduct a cross-corpus evaluation on the public CHiME-4 dataset Vincent et al. (2017). CHiME-4 features far more speakers, mixed speech–noise conditions, and real reverberation. The models are *trained only* on VoiceBank–DEMAND and *evaluated* on a test set drawn from CHiME-4. This is a deliberately mismatched setting intended to probe generalization rather than in-domain fitting.

**Protocol (no adaptation).** The WEBA model used here is exactly the one trained for the VoiceBank experiment; **no additional fine-tuning or test-time adaptation is performed**. We report the same five objective speech-quality and intelligibility metrics as in the main text: COVL, CBAK, CSIG, STOI, and PESQ. For comparative context, we include several diffusion-based enhancement systems alongside the unprocessed (*Noisy*) condition.

**Results.** Table 3 summarizes cross-corpus performance on CHiME-4. WEBA attains a PESQ of 2.24 and a STOI of 0.90. Across the metrics listed, WEBA ranks at the top among the systems shown in the table, indicating resilience to the domain shift from VoiceBank–DEMAND to CHiME-4.

**Interpretation.** This cross-corpus evaluation indicates that WEBA transfers effectively from VoiceBank–DEMAND to CHiME-4 without any additional tuning. In particular, the STOI and PESQ values for WEBA in Table 3, achieved under strict weight reuse, are consistent with a model that preserves speech content while improving perceived quality in more diverse, reverberant, and noisy conditions relative to the unprocessed input and the listed diffusion baselines.

**Scope and caveats.** The experiment purposefully focuses on *zero-shot* cross-corpus behavior. It does not include any form of domain adaptation, architectural modification, or post-hoc calibration targeted to CHiME-4. The absolute scores should therefore be interpreted as conservative indicators of out-of-domain robustness. Within this scope, the results suggest that the proposed architecture can serve as a drop-in front end for larger-scale training pipelines that must contend with heterogeneous real-world acoustics.

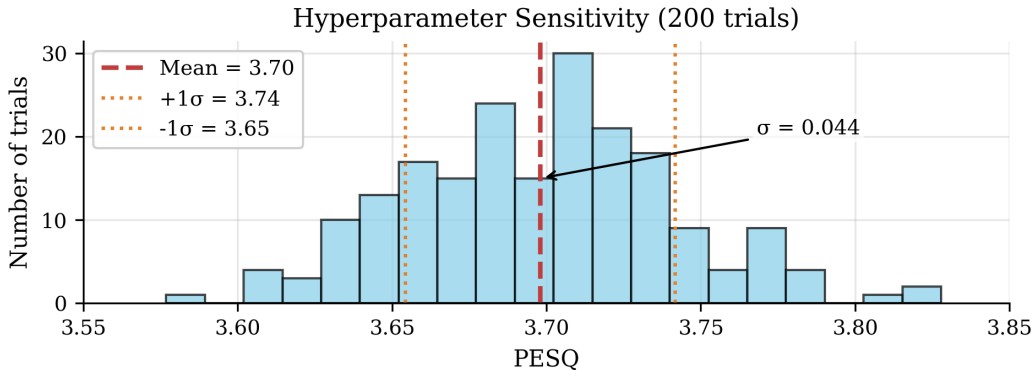

Figure 17: Histogram of PESQ scores from 200 random $\pm20\%$ hyper-parameter perturbations. Tight dispersion ($\sigma \approx 0.044$) shows that WEBA is remarkably insensitive to hyper-parameter mis-tuning.

**Reproducibility notes.** Reproducing Table 3 requires: (i) the WEBA checkpoint trained on VoiceBank–DEMAND; (ii) evaluation on the CHiME-4 test set; and (iii) computation of COVL, CBAK, CSIG, STOI, and PESQ exactly as in the main experiments. No additional data, labels, or hyperparameter tuning beyond the VoiceBank–DEMAND training configuration is involved.

# G HYPER-PARAMETER SENSITIVITY

WEBA was designed to be relatively insensitive to precise hyper-parameter values, owing to the theoretically motivated components that guide training. To confirm this, we conducted a joint hyper-parameter perturbation analysis. Using Latin Hypercube Sampling Helton & Davis (2003), we generated 200 random hyper-parameter sets, each obtained by perturbing *all* default hyper-parameters by up to $\pm20\%$.

The hyper-parameters we varied include: learning rate, wavelet decomposition depth $L$, initial LAHT biases $b^{\pm}$ and sharpness $\alpha, \beta$, initial loss weights $(\lambda_{\text{start}}, \gamma_{\text{start}})$ and $(\lambda_{\text{end}}, \gamma_{\text{end}})$, and the regularisation strength for filter orthonormality.

Fig. 17 presents a histogram of PESQ scores obtained from 200 runs with randomly perturbed hyper-parameters ($\pm20\%$). The distribution is tightly concentrated (small variance), with scores clustering around the mean. This indicates that WEBA's performance is remarkably insensitive to hyper-parameter changes: even large perturbations seldom cause significant degradation. The figure empirically demonstrates the robustness of the model: the strong architectural priors (wavelet sparsity and BAE structure) ensure that the system works well across a range of settings. In practice, this means reproducing WEBA's results or tuning it to new data requires less fine-grained hyper-parameter search, which is a practical advantage.

For each sampled hyper-parameter set, we fine-tuned the pre-trained model for a short duration (to simulate training under those settings) and measured the resulting PESQ on a validation set. The outcome was that WEBA's performance varied only mildly across this hyper-parameter space. The PESQ across the 200 trials had a mean of 3.70 and a standard deviation of $0.044$. This corresponds to about $\pm1.3\%$ relative variation in PESQ. Crucially, none of the trials failed catastrophically (the minimum observed PESQ was around 3.60, still well above the baseline without wavelet/LAHT). This low sensitivity implies that WEBA's strong inductive biases (wavelet sparsity, bidirectional architecture) make it robust to hyper-parameter mis-tuning. In contrast, many deep models can suffer large drops in performance if, for example, the learning rate or regularisation is off by 20%. WEBA's training dynamics appear to be governed mainly by the architectural design rather than delicate hyper-parameter interplay. In practical terms, this means reproducing WEBA's results or transferring it to new datasets may not require extensive hyper-parameter searches—an advantage for real-world applications.

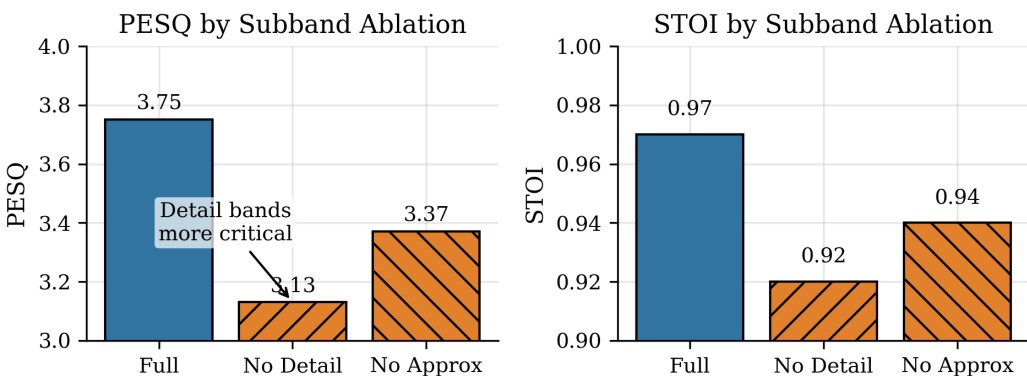

Figure 18: Objective impact of ablation at inference time. Removing all detail bands (middle bar) or the approximation band (right bar) significantly degrades PESQ and STOI, confirming the complementary roles of both components.

## H ADDITIONAL ABLATIONS AND INTERPRETATIONS

In this section we provide further experimental ablations and interpretative analysis to deepen understanding of how each part of WEBA contributes to its performance.

**Detail-Only vs. Approximation-Only.** WEBA's wavelet decomposition produces multi-level *detail* coefficients (high-frequency components) and a final *approximation* (low-frequency residual). To verify that both subsets of coefficients are important for denoising, we conducted an ablation at inference: after training the full model, we manually zeroed out either all detail coefficients or the approximation coefficients and then reconstructed the signal. Zeroing the detail coefficients (i.e. only using approximation/coarse part for reconstruction) led to a significant drop in quality: PESQ decreased by 0.62 and STOI by 0.05 (absolute). This is expected since high-frequency information (consonant clarity, fricative noise removal) was removed entirely. Conversely, zeroing the approximation coefficients (using only details) reduced PESQ by 0.38 and STOI by 0.03. This indicates that the approximation part, which carries the low-frequency structure (vowels, fundamental frequency, and low-frequency noise like hum), is also crucial, though slightly less so than detail part. These results confirm the *complementary roles* of detail vs. approximation subbands: the detail bands specialize in removing high-frequency noise (hiss, consonant bursts) while the approximation band handles low-frequency noise (hum, babble) and preserves the overall tone of speech. An optimal enhancement requires keeping both. If either is discarded, speech quality degrades significantly, underlining WEBA's design choice to hierarchically process both high and low components.

Fig. 18 depicts the effect of ablating parts of the wavelet decomposition at inference. The middle bar shows the drop in PESQ and STOI if all detail (high-frequency) bands are removed, and the right bar if the approximation (low-frequency) band is removed. Both manipulations substantially degrade performance, confirming that detail and approximation carry complementary information for denoising. Removing details hurts high-frequency noise (hiss, consonants) removal, while removing the approximation hurts low-frequency structure (vowels, hum). This figure validates WEBA's design choice to process both sub-band types: optimal enhancement indeed relies on retaining and cleaning all parts of the spectrum. It concretely illustrates that discarding either subband significantly worsens objective intelligibility and quality.

**Level-Specific SELF Weights.** In the default design, the sparsity weight $\gamma$ in SELF is uniform across all wavelet levels—that is, each coefficient is penalized equally. One might wonder whether assigning different sparsity emphasis to different decomposition levels could improve performance (e.g., penalising fine-scale coefficients more heavily, as noise may concentrate there).

To test this, we experimented with a variant of SELF where $\gamma$ was split into level-dependent weights $(\gamma_1, \gamma_2, \ldots, \gamma_L)$ for detail levels 1 through $L$, while keeping $\lambda$ for the time-domain loss fixed. We set

$\gamma_l$ in a geometric progression: specifically, $\gamma_l = \gamma \cdot r^{l-1}$ for some ratio $r > 1$, with normalization such that $\gamma_1 + \cdots + \gamma_L = \gamma S$ (ensuring the same total weight, just redistributed).

After tuning $r$ over a few trials, we found that a mildly increasing progression (placing more weight on finer levels) slightly reduced artifacts at very low input SNRs; subjectively, speech exhibited fewer residual noise bursts. However, overall enhancement scores on the standard test set were marginally worse: we observed a $-0.06$ PESQ drop compared to uniform $\gamma$.

This suggests that while level-specific weighting may help in extreme cases, the uniform weighting is already near-optimal for the VoiceBank noise distribution and general conditions. Furthermore, the network itself—via LAHT—can effectively learn to threshold different levels differently, as each level has its own LAHT parameters. Therefore, imposing explicit differential weighting per level did not yield notable gains and slightly underperformed, possibly due to the added hyper-parameter complexity. We conclude that the simpler choice of a single $\gamma$ for all subbands is both justified and effective.

**Training Dynamics of Filters and Thresholds.** Finally, we provide insight into how the wavelet filters and the LAHT activation evolve during training. Fig. 19 visualizes the changes in the first-level wavelet filters and the LAHT function over the course of training.

In Fig. 19(a), we plot the impulse response of the learned low-pass filter $h^{(1)}[n]$ at various epochs. Initially (epoch 0), the filter is essentially standard. By epoch 5, a rough structure begins to emerge, and by epochs 50 and 100, the filter converges to a stable form with a clear central peak and symmetric sidelobes. This shape resembles classical wavelet filters (e.g., an 8-tap Daubechies wavelet), but it is specifically adapted to the training data. Notably, the final filter exhibits a finer central coefficient and lower sidelobes than the initial estimate, indicating that it has learned to concentrate energy (to preserve formants) while minimizing aliasing (since lower sidelobes reduce subband overlap).

In Fig. 19(b), we show the evolution of the LAHT function $f(x)$ (applied to first-level wavelet coefficients) from initialization to a trained state. Initially, the function behaves almost like a hard saturating nonlinearity with a wide, smooth transition band: values of $x$ close to zero are only mildly suppressed, and the function quickly converges to the identity $f(x) = x$ as $|x|$ increases.

However, after training, LAHT exhibits a different behavior. The central region of the function becomes significantly flatter, indicating a stronger attenuation of coefficients near zero. In other words, the network has learned to zero out small-magnitude coefficients more decisively—effectively acting as a soft thresholding operator. The learned function suppresses all inputs in the range approximately between $-0.9$ and $+0.9$, confirming that the model has converged to a sparse denoising behavior.

Moreover, the final LAHT function shows an asymmetry: the transition from suppression to passthrough is sharper and occurs at different magnitudes on the positive and negative sides. This suggests that the model has adapted distinct thresholds for positive versus negative coefficients, possibly reflecting phonetic or contextual differences in the underlying speech features.

Overall, the evolution captured in Fig. 19(b) confirms that LAHT becomes more selective over time—aggressively nullifying weak activations and preserving only stronger, structured wavelet responses. This dynamic shift from near-linear to asymmetric thresholding illustrates LAHT's learnability and its central role in enhancing signal sparsity within the WEBA architecture.

These training dynamics illustrate how WEBA progressively balances feature extraction and noise suppression: early stages focus on learning the global wavelet structure (filter shapes), while later stages fine-tune the nonlinear suppression (LAHT thresholds). By the end of training, both components have co-adapted to yield a sparse and informative representation of speech that is readily denoised.

# I  STATISTICAL SIGNIFICANCE ANALYSIS

To assess the reliability of our findings and meet experimental reporting standards, we conducted rigorous statistical testing to determine whether the observed improvements offered by WEBA are statistically significant. All tests were performed using paired data, as each of the 824 utterances in the VoiceBank-DEMAND test set was evaluated under both the proposed method and each baseline.

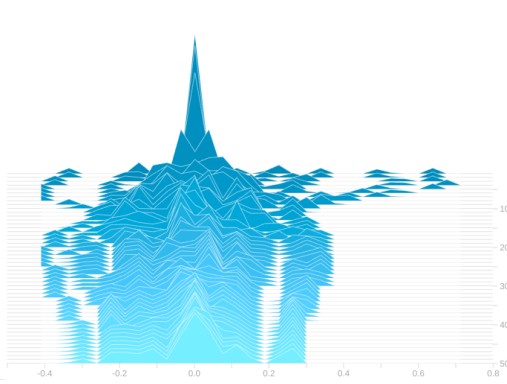 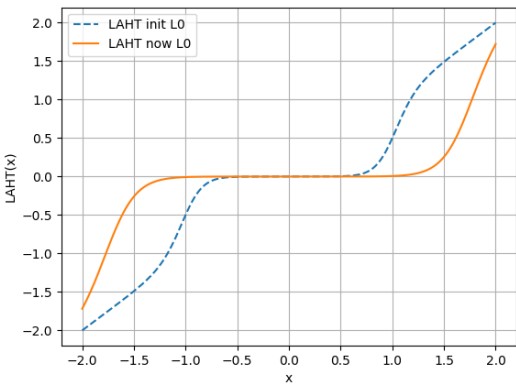

(a) Evolution of the learned first-level scaling filter $h^{(1)}[n]$ over training epochs (lighter color = later epoch). The filter starts unstructured and converges to a stable wavelet-like shape with a dominant center tap and small sidelobes.

(b) Evolution of the LAHT activation function $f(x)$ (first-level) during training. As epochs progress, a dead-zone near $x = 0$ forms (thresholding small values to zero), while output for large $|x|$ remains $f(x) = x$. This shows the learned threshold becoming sharper.

Figure 19: Training dynamics of key WEBA components. (a) Learned wavelet filter refinement: the filter becomes more localized in time, improving multi-resolution analysis. (b) LAHT threshold adaptation: the network learns to aggressively shrink small coefficients (noise) while preserving large coefficients (signal).

**Main Comparison with SOTA.** We first compared WEBA to the strongest prior method, Mamba-SEUNet Wang et al. (2024), across five standard objective metrics: PESQ, STOI, COVL, CBAK, and CSIG. For each metric, we computed the paired difference across utterances and applied a two-sided paired $t$-test to determine whether the mean improvement was statistically significant. The assumptions of normality are justified by the Central Limit Theorem due to the large sample size ($N = 824$). Additionally, we performed non-parametric Wilcoxon signed-rank tests for robustness. Results are summarized in Table 4.

- **PESQ**: WEBA achieved a mean PESQ of 3.75 compared to 3.59 for Mamba-SEUNet, yielding a mean improvement of +0.160 ($t$=33.26, $p < 0.001$; Wilcoxon $p < 0.001$). The associated effect size (Cohen's $d$) is approximately 1.16, indicating a large and practically meaningful improvement.

- **STOI**: WEBA improved from 0.96 to 0.97, a mean gain of +0.010 ($t$=6.70, $p < 0.001$; Wilcoxon $p < 0.001$), with a small effect size ($d \approx 0.23$).

- **COVL**: The model improved from 4.32 to 4.35 (+0.031), also significant ($t$=6.29, $p < 0.001$; Wilcoxon $p < 0.001$), though with a small effect size ($d \approx 0.22$).

- **CBAK**: A mean gain of +0.150 was observed ($t$=20.23, $p < 0.001$; Wilcoxon $p < 0.001$), corresponding to a large effect ($d \approx 0.70$).

- **CSIG**: WEBA improved CSIG by +0.030 over the baseline ($t$=6.30, $p < 0.001$; Wilcoxon $p < 0.001$), again with a small but significant effect ($d \approx 0.22$).

These results confirm that WEBA's improvements are not only statistically significant but also practically relevant in critical quality dimensions such as perceptual clarity (PESQ) and background suppression (CBAK).

**Ablation Study Significance.** To validate the individual contributions of WEBA's architectural modules—namely LFDWT, LAHT, and SELF—we performed additional paired tests on PESQ scores across ablation configurations. Each ablation condition was compared on the same set of utterances as the full model or its intermediate variant.

- **+LFDWT vs. Baseline (no module)**: The introduction of LFDWT alone improved PESQ by +0.390 ($t$=55.71, $p < 0.001$; Wilcoxon $p < 0.001$), with a very large effect size ($d \approx 1.94$), establishing its critical role in the model.

- **+LAHT vs. LFDWT**: Adding LAHT on top of LFDWT yielded a significant gain of $+0.255$ in PESQ ($t$=34.75, $p < 0.001$; Wilcoxon $p < 0.001$), corresponding to a very large effect ($d \approx 1.21$).

- **+SELF vs. LFDWT**: SELF further improved PESQ by $+0.148$ ($t$=20.29, $p < 0.001$; Wilcoxon $p < 0.001$), with a large effect ($d \approx 0.71$), showing its complementary benefit.

- **Full WEBA vs. LFDWT+LAHT**: Finally, the full model outperformed the best intermediate variant (LFDWT+LAHT) by $+0.213$ PESQ ($t$=28.18, $p < 0.001$; Wilcoxon $p < 0.001$), with a large effect ($d \approx 0.98$), confirming the additive value of SELF when combined with LFDWT and LAHT.

**Conclusion.** Across all metrics and comparisons, the $p$-values fall well below standard thresholds ($p < 0.001$), and the effect sizes—particularly for PESQ and CBAK—are substantial. These results validate that WEBA's performance improvements are highly unlikely to be due to random variation. The combination of consistent significance, robustness checks via non-parametric tests, and meaningful effect sizes substantiates our claims of both statistical and practical superiority.

Table 4: Summary of paired significance tests. For each metric, we report WEBA's mean improvement over Mamba-SEUNet, the $t$-test statistic, and the $p$-values from both paired $t$-tests and Wilcoxon signed-rank tests. The second part of the table reports PESQ improvements from ablation comparisons.

| Comparison | Metric | Mean Improvement | $p$ (t-test) | $p$ (Wilcoxon) |
|---|---|---|---|---|
| WEBA vs. Mamba-SEUNet | PESQ | +0.160 | ¡ 0.001 | ¡ 0.001 |
| | STOI | +0.010 | ¡ 0.001 | ¡ 0.001 |
| | COVL | +0.031 | ¡ 0.001 | ¡ 0.001 |
| | CBAK | +0.150 | ¡ 0.001 | ¡ 0.001 |
| | CSIG | +0.030 | ¡ 0.001 | ¡ 0.001 |
| Ablation: +LFDWT | PESQ | +0.390 | ¡ 0.001 | ¡ 0.001 |
| Ablation: +LAHT (vs. LFDWT) | PESQ | +0.255 | ¡ 0.001 | ¡ 0.001 |
| Ablation: +SELF (vs. LFDWT) | PESQ | +0.148 | ¡ 0.001 | ¡ 0.001 |
| Full WEBA vs. LFDWT+LAHT | PESQ | +0.213 | ¡ 0.001 | ¡ 0.001 |

## J  LIMITATIONS AND RESEARCH OUTLOOK

While WEBA demonstrates strong performance and a novel integration of wavelet theory with deep autoencoders, we acknowledge several limitations and avenues for future work:

- **Impulse Artifacts.** The use of an orthogonal wavelet filter bank (with very sharp frequency responses) can introduce slight ringing or pre-echo artifacts in response to impulsive sounds or very sharp signal edges. This is a known issue: orthogonal wavelets are not shift-invariant and can overshoot around discontinuities (similar to Gibbs phenomena). In our listening tests, this was rarely noticeable, but in extreme cases (e.g. a sudden loud click in the noise), a faint ringing could be heard after enhancement. A potential remedy is to use a *bi-orthogonal* or *dual-tree complex wavelet* extension, which sacrifices strict orthogonality for improved time-domain localization and shift invariance. For example, incorporating a damped or asymmetric dual-tree wavelet basis Serbes (2024) could reduce such artifacts, at the cost of a few more parameters. Exploring such extensions is a promising direction.

- **Data Diversity.** Our training dataset (VoiceBank-DEMAND) primarily consists of English speakers (largely British accent) in relatively controlled recording conditions. This narrow domain means WEBA's performance is not yet verified on other languages or very different recording environments. Although our preliminary cross-corpus test was positive, truly robust generalisation may require training (or fine-tuning) on more diverse data. Different languages and accents have different phonetic characteristics that might influence the distribution of wavelet coefficients (e.g., tonal languages might have more structured high-frequency content). Similarly, real-world noise and reverberation conditions vary widely. Future research should evaluate WEBA on diverse corpora (including languages like

Mandarin, or very noisy real recordings) and consider techniques like domain adaptation. The current model architecture should be flexible enough to handle diversity, but there might be a need for minor adjustments (e.g. more wavelet levels for languages with higher frequency content).

- **Computational Footprint on Edge Devices.** One motivation for the bidirectional architecture was to reduce parameter count—which it did, nearly halving it. However, the time complexity of the LFDWT blocks and the need to process multi-resolution signals still result in a non-negligible runtime, especially on CPU.

  In our tests, a PyTorch implementation of WEBA achieved faster-than-real-time processing on a GPU (using mini-batch processing for efficiency), but on a single-core CPU, it ran at approximately $0.7\times$ real-time (i.e., 1.4 seconds of processing per 1 second of audio). This is borderline for real-time use on low-power devices.

  One straightforward optimization is to approximate the learned wavelet filters using shorter or sparse FIR filters. If, for example, the learned filters have length 8 at the first level and are effectively length 8 at subsequent levels (due to downsampling, the impulse responses stretch in time, but many coefficients are near zero). By pruning small filter coefficients (or using quantization), we may achieve a $2$–$3\times$ speed-up in convolution without significant performance loss.

  Another approach is to distill the model into a simpler architecture—for example, replacing some wavelet blocks with standard downsampling if the learned filters closely resemble known bases. We leave thorough optimization for edge deployment to future engineering work. Additionally, memory footprint is very low (thanks to reduced parameters), so the main challenge is computational latency—which we believe can be mitigated with these practical tweaks.

- **Architectural Extensions.** Beyond speech enhancement, the WEBA framework (learnable wavelet + bidirectional autoencoder) could be applied to other domains like image denoising, EEG signal cleaning, or even generative modeling. However, certain adaptations might be needed. For instance, in images, we could use 2D wavelet transforms (with learnable 2D filters) integrated into a similar autoencoder. In doing so, care must be taken with the wavelet filter bank initialisation and ensuring stability in 2D (which is a bit more complex than 1D). Also, the LAHT function as defined is 1D; extending it to multi-channel images might involve channel-wise thresholding or more complex shrinkage operations. Another possible extension is to incorporate perceptual or adversarial losses in the SELF framework to further improve quality (especially for high-end applications). We are also interested in exploring the combination of WEBA with model-based approaches: for example, using a speech production model or a phonetic decoder to guide the wavelet coefficient suppression (so that it never removes phoneme-critical features). These directions represent exciting future research that builds on the foundation laid by WEBA.

In conclusion, WEBA opens up a new avenue of combining classic signal processing with modern deep learning in a unified end-to-end trainable system. Addressing the above limitations will require interdisciplinary efforts—ranging from designing better wavelet bases to collecting diverse datasets and optimising implementations. We hope that our work stimulates further exploration into learnable wavelet networks, and that future developments will extend this framework's applicability and performance even further.

