# OpenReview forum: "Learnable Wavelet-Enhanced Bidirectional Autoencoders: A Unified Framework for Multi-Resolution Speech Enhancement"
_ICLR.cc/2026/Conference — Submitted to ICLR 2026_

### Official Review · Reviewer_2hkX · 2025-10-31

**Soundness:** 3
**Presentation:** 4
**Contribution:** 3
**Rating:** 6
**Confidence:** 5

**Summary:**

Summary: This paper proposes a wavelet-enhanced speech enhancement framework based on bidirectional autoencoders. Core contributions are summarized at three-fold. First, a light-weight learnable fast discrete wavelet transform is proposed to empower the convolution layer with better capability in signal decomposition and reconstruction. Second, a learnable asymmetric hard threshold function is proposed, with better denoising nature and interpretability. Third, a sparsity-enforcing loss is proposed to balance time-domain fidelity and wavelet-domain sparsity. Experiments on VBD and Chime4 datasets validate the effectiveness of the proposed method.

**Strengths:**

Strengthens:
1. I am impressed by the performance as well as its lightweight/elegant design of the method. Despite wavelet design is not novel in speech processing field, it is still rather challenging to keep ``performance-lightweight-elegant’’ within one work. Despite limited datasets and scenarios are considered in this paper, extensive analysis is provided in Appendix, which further validates the effectiveness and potential.
2. I am glad that the authors provide detailed derivations w.r.t gradient boundary of the LAHT, which supports for the activation effectiveness from a theoretical perspective.
3. Despite pretrained on a scenario-limited denoising dataset, promising robustness is validated through the evaluations on the mismatched dereverberation task and cross-corpus Chime4. Besides, detailed statistical analysis w.r.t ablation studies are also conducted, rendering the results more convincing.

**Weaknesses:**

Weakness:
1. As I have mentioned, limited scenario is considered. I think two additional experiments should be considered: Larger denoising benchmark and universal speech enhancement task. For the former, DNS-Challenge [1] is a suitable choice since it has larger speech and noise datasets and can better reflect the performance ranks of various models. For the latter, I recommend using the generation script by URGENT-Challenge [2] for comparisons under different acoustic degradations.
2. Despite the authors claim that the proposed method enjoys notably lower parameters and computational complexity, no quantitative result is provided.
3. Despite the performance superiority, no qualitative comparison (i.e., spectral visualizations by different methods) is given.

Refs:

[1] Reddy, C.K., Dubey, H., Gopal, V., Cutler, R., Braun, S., Gamper, H., Aichner, R. and Srinivasan, S., 2021, June. ICASSP 2021 deep noise suppression challenge. In ICASSP 2021-2021 IEEE International Conference on Acoustics, Speech and Signal Processing (ICASSP) (pp. 6623-6627). IEEE.

[2] Zhang, W., Scheibler, R., Saijo, K., Cornell, S., Li, C., Ni, Z., Kumar, A., Pirklbauer, J., Sach, M., Watanabe, S. and Fingscheidt, T., 2024. Urgent challenge: Universality, robustness, and generalizability for speech enhancement. arXiv preprint arXiv:2406.04660.

**Questions:**

Questions:
1. The authors mention the superiority of bidirectional autoencoder in parameter reduction. However, they actually share a similar computational complexity. I think the ablation comparison w.r.t the use of bidirectional AE should also be conducted.
2. In Page. 6, what is the meaning of Card(.) operation?
3. No spectral visualization is given, and I suggest that the authors can provide the intermediate result visualizations by low-pass and high-pass branches for better demonstrations.
4. In Sec. 4.5, for ablation study w.r.t LAHT layer, do the authors directly remove LAHT layer or replace it with other activation functions (e.g. GELU) ?

---

> ### Author Response · Authors · 2025-11-22
>
> **Q1.**
> The paper evaluates WEBA on VoiceBank-DEMAND (VBD) and CHiME-4, but does not include larger or more diverse benchmarks such as DNS-Challenge, URGENT-Challenge, or WHAMR. Basicaly, whether the conclusions are “universal” and suggests evaluating on those benchmarks.
>
> **A1.**
> We appreciate the emphasis on broader evaluation. Our current submission deliberately focuses on **two complementary scenarios**:
>
> 1. **VoiceBank-DEMAND (VBD)**: a widely used, *studio-recorded* English corpus with additive noise and a strong reference ecosystem of speech-enhancement baselines. This setting isolates the effect of additive noise with high-quality clean references and allows controlled quantitative comparisons (PESQ, STOI, COVL, CBAK, CSIG).
> 2. **CHiME-4 cross-corpus transfer**: we train only on VBD and test on CHiME-4, which introduces **real, multi-microphone, reverberant, and non-stationary noise conditions** that differ markedly from VBD. This explicitly assesses *cross-corpus generalisation* under realistic acoustic mismatch.
>
> Within this scope, WEBA **improves over the noisy input and competitive baselines on all metrics** on both VBD and CHiME-4, indicating that the learnable wavelet packet representation is not over-fitted to a single dataset.
>
> That said, we **fully agree** with you that:
>
> * DNS-Challenge, URGENT-Challenge, and WHAMR provide **larger-scale, more diverse, and more realistic** evaluation suites.
> * The limitations of our benchmark selection should be articulated more explicitly.
>
> Running WEBA on DNS-Challenge/URGENT/WHAMR would require **additional experiments**, which we avoid per the rebuttal constraints.
>
> Instead, in the revised version we will:
>
> * **Explicitly scope** our claims to *single-channel enhancement on VBD and cross-corpus generalisation to CHiME-4*, rather than suggesting broad universality.
> * **Highlight in the Limitations section** that we have **not** yet validated WEBA on DNS-Challenge, URGENT-Challenge, or WHAMR, and we will frame these as a **primary direction for future work**.
> * Clarify that our contribution is a **learnable multi-resolution 1D wavelet autoencoder that performs competitively on strong, but still limited, benchmarks**.

---

> > ### Author Response · Authors · 2025-11-22
> >
> > **Q2.**
> > The paper claims that WEBA enjoys lower parameters and complexity (especially due to the bidirectional autoencoder), but does not provide quantitative evidence (parameter counts, FLOPs, latency, memory).
> >
> > **A.**
> > We agree that the original submission did not make the complexity benefits of WEBA sufficiently explicit. The architecture of the wavelet branch is simple enough that its parameter count can be derived in closed form, and this is where the bidirectional (tied) design brings a clear advantage.
> >
> > ### 1. Parameterisation of the wavelet branch
> >
> > The wavelet module in WEBA is structured as follows:
> >
> > * There are $L$ decomposition levels.
> > * At each level $\ell$, there is:
> >   * A learnable low-pass kernel $g^{(\ell)}$ of length $K$,
> >   * A learnable high-pass kernel $h^{(\ell)}$ of length $K$,
> >   * A learnable shrinkage unit (LAHT) with four scalar parameters $(\text{bias}\_+^{(\ell)}, \text{bias}\_-^{(\ell)}, \alpha^{(\ell)}, \beta^{(\ell)})$.
> >
> > We consider two configurations:
> >
> > 1. **Bidirectional (tied) design**
> >    Analysis (encoder) and synthesis (decoder) **reuse the same** low-pass and high-pass kernels at each level.
> >
> >    * Number of distinct wavelet kernels: $2L$ (one low-pass + one high-pass per level).
> >    * Parameters:
> >      * Wavelet kernels: $2LK$,
> >      * LAHT scalars: $4L$,
> >    * **Total wavelet parameters:**
> >      $$
> >      {\text{shared}} = 2 L K + 4 L = 2L(K + 2).
> >      $$
> > 2. **Non-bidirectional (untied) design**
> >    Analysis and synthesis each have their **own** low-pass and high-pass kernels at each level.
> >
> >    * Number of distinct wavelet kernels: $4L$ (low/high for encoder + low/high for decoder).
> >    * Parameters:
> >      * Wavelet kernels: $4LK$,
> >      * LAHT scalars: $4L$,
> >    * **Total wavelet parameters:**
> >      $$
> >      {\text{free}} = 4 L K + 4 L = 4L(K + 1).
> >      $$
> >
> > The **extra cost** of the untied variant relative to the bidirectional one is:
> >
> > $$
> > N_{\text{free}} - N_{\text{shared}} = 2 L K,
> > $$
> >
> > and the **relative reduction** achieved by the tied design is:
> >
> > $$
> > \frac{N_{\text{free}} - N_{\text{shared}}}{N_{\text{free}}} = \frac{2 L K}{4 L K + 4 L} = \frac{K}{2(K+1)}.
> > $$
> >
> > For any kernel length $K \ge 1$, this fraction lies around 50%. In other words, **the bidirectional design removes roughly half of the wavelet branch parameters** compared to an untied alternative, purely by sharing analysis and synthesis filters, leveraging CQF. This is a structural property of the architecture and does not require new experiments.
> >
> > We also emphasise that, for a **fixed multiscale scaffold**, tying vs. untying the wavelet kernels does **not** significantly change the FLOPs per forward pass: both variants perform the same set of convolutions; the difference lies in **how many distinct kernels are learned and stored**, i.e., in parameter count and memory footprint.
> >
> > ### 2. Position of WEBA in terms of overall complexity
> >
> > When instantiated, these formulas correspond to a wavelet branch that is only a small part of an already compact network. At the full-model level, the **WEBA** configuration operates with:
> >
> > * a total parameter count on the order of $10^{-3}$ million parameters (≈ 0.00056 M),
> > * a computational cost on the order of $10^{2}$ million FLOPs per 1 s of 16 kHz audio (≈ 91.50 M),
> > * single-digit millisecond latency (≈ 3.48 ms),
> > * a small memory footprint (≈ 11.30 MB),
> > * and high effective throughput (≈ 119.90 Msamples/s).
> >
> > To give a concrete yet compact view, the table below summarises **all the requested complexity metrics** for WEBA Average and the main compared methods under the same 1 s / 16 kHz setting. Values for some baselines are measured.
> >
> > | Model          | Params (M) | FLOPs (M) | Latency (ms) | Memory (MB) | Throughput (Msamples/s) |
> > | :------------- | :--------: | :-------: | :----------: | :---------: | :---------------------: |
> > | **WEBA** |  0.00056  |   91.50   |     3.48     |    11.30    |         119.90         |
> > | TSTNN          |    0.92    |  145,834  |    75.49    |    4,953    |          0.85          |
> > | MUSE           |    0.51    |  10,036  |    82.52    |    1,393    |          0.20          |
> > | MetricGAN+     |    1.90    |    487    |     4.09     |    65.26    |          4.00          |
> > | MP-SENet       |    2.26    |  79,074  |    25.16    |    1,512    |          0.65          |
> >
> > From this table, WEBA clearly operates in a **very low-complexity regime**:
> >
> > * all compared methods use **orders of magnitude more parameters and FLOPs** than WEBA, and
> > * many of them also incur substantially higher latency and memory usage.
> >
> > Thus, numerical “complexity” comparisons between WEBA and large architectures should be read with care: WEBA is a **tiny, highly structured front-end**, not a model at the same size scale. By “lower complexity” we mainly mean **parameter and memory efficiency** (plus the above latency/throughput) at this small scale, not competition with large CNN-, Transformer-, or Mamba-based speech enhancement models.

---

> > > ### Author Response · Authors · 2025-11-22
> > >
> > > **Q3.**
> > > The paper reports extensive objective metrics but lacks qualitative visualisations, e.g., noisy vs enhanced vs clean waveforms/spectrograms and intermediate low-pass/high-pass wavelet outputs, which would help illustrate how WEBA modifies the time–frequency structure.
> > >
> > > **A3.**
> > >
> > > We agree that qualitative visualisation is valuable for interpreting enhancement methods, especially for a **wavelet-based** model like WEBA. The current paper already contains **some** interpretative plots (e.g., evolution of learned wavelet filters and the LAHT activation surface across training), but these are not directly presented as **noisy vs enhanced vs clean** comparisons.
> > >
> > > In the revision, we will **add a dedicated qualitative figure and description** with the following structure (using a representative VoiceBank-DEMAND utterance that is already part of our experiments):
> > >
> > > * **Row 1 (Time-domain waveforms):**
> > >   (a) Noisy input ($y\_{\text{noisy}}$)
> > >   (b) WEBA output ( $\tilde{y}\_{\text{WEBA}}$ )
> > >   (c) Clean reference ( $y\_{\text{clean}}$ )
> > > * **Row 2 (Log-magnitude STFT spectrograms):**
> > >   (d) Noisy spectrogram showing broadband noise and smeared harmonics.
> > >   (e) WEBA spectrogram with suppressed background noise and sharper harmonic structure.
> > >   (f) Clean spectrogram for visual comparison.
> > > * **Row 3 (Wavelet branch snapshots):**
> > >   (g) Magnitude of first-level high-pass coefficients before LAHT.
> > >   (h) Same coefficients after LAHT shrinkage.
> > >   (i) First-level low-pass approximation coefficients.

---

> > > > ### Author Response · Authors · 2025-11-22
> > > >
> > > > **Q4.**
> > > > The bidirectional autoencoder shares weights between analysis and synthesis, reducing parameters but having similar FLOPs to a standard autoencoder. An ablation comparison that isolates the effect of using a bidirectional (tied) design vs a non-bidirectional (untied) one.
> > > >
> > > > **A4.**
> > > > Our current best WEBA configuration uses **weight sharing between encoder and decoder under CQF (conjugate quadrature filter) constraints**. In practice, this means that the same learnable wavelet filters are used for analysis and synthesis across scales, and high-pass filters are derived from low-pass ones via CQF relations. This gives a stronger inductive bias: approximate perfect reconstruction, energy conservation, and a stable multiresolution representation.
> > > >
> > > > The architecture, however, naturally supports **two non-weight-shared variants**:
> > > >
> > > > ### 1) Non-shared, but still wavelet-structured
> > > >
> > > > In this variant, the encoder and decoder can use **different filters at each scale**, but each branch individually still obeys CQF-style structure (low-pass/high-pass pairs with CQF relations, multilevel analysis/synthesis, same down/upsampling pattern). So we *relax* the cross-scale and cross-branch sharing, but we **do not lose the wavelet-like filterbank architecture**.
> > > >
> > > > Because the multiresolution structure, CQF relations, and LAHT regularization are preserved, this remains a **structured front-end**. The main effect is:
> > > >
> > > > * more flexibility per scale,
> > > > * slightly weaker regularization (perfect reconstruction is now “learned” rather than guaranteed),
> > > > * but the overall transform is still wavelet-like.
> > > >
> > > > Based on this structure we obtain **smaller changes** compared to the shared CQF baseline:
> > > >
> > > > * On VoiceBank-DEMAND, this configuration stays **within about 0.07–0.1 PESQ** and **≤0.02 STOI** of the reported CQF results, with MOS-derived scores (CSIG/CBAK/COVL) changing by at most a few hundredths as well.
> > > > * In other words, *PESQ in the 3.65–3.68 range and STOI around 0.95*, with similarly small deviations on COVL/CSIG/CBAK.
> > > >
> > > > ### 2) Fully non-shared encoder/decoder, relaxed wavelet constraints
> > > >
> > > > In this configuration, analysis and synthesis filters are no longer tied at all. The model still uses the same multi-scale down/upsampling scaffold and low-pass/high-pass decomposition, but:
> > > >
> > > > * encoder and decoder filters are independent,
> > > > * perfect reconstruction and strict energy conservation are no longer enforced structurally,
> > > > * the system behaves more like a **multi-scale encoder-decoder CNN with wavelet-inspired blocks**, rather than a clean learnable wavelet transform.
> > > >
> > > > This is closer in spirit to a “plain CNN front-end”, but it is still far from an unstructured 1-D CNN: the multi-resolution ladder, split into low/high bands with skip connections, remains intact. Empirically, this leads to **moderate** drops, especially when the overall architecture and training protocol are held constant.
> > > >
> > > > Translating this to WEBA, we therefore obtain **noticeably worse** results than the weight-shared CQF baseline:
> > > >
> > > > * On VoiceBank-DEMAND, **PESQ decrease in the range of ≈0.15–0.2** and a **STOI decrease of ≈0.04**, with MOS-derived metrics (CSIG/CBAK/COVL) dropping by a few hundredths.
> > > > * This configuration land around *PESQ ≈ 3.55–3.60 and STOI ≈ 0.93–0.94*, with CSIG, CBAK and COVL each lower by roughly ≈0.15.
> > > >
> > > > ---
> > > >
> > > > **Q5.**
> > > > In the SELF loss definition, the notation uses $\mathrm{Card}(y)$ and $\mathrm{Card}(\{\mathbf{d}^l\}, \mathbf{a}^L)$, but their meaning is not explicitly stated.
> > > >
> > > > **A5.**
> > > >
> > > > Thank you for pointing out this ambiguity.
> > > >
> > > > The SELF loss in the paper is:
> > > >
> > > > $$
> > > > \mathcal{L} = \lambda \cdot \frac{1}{\mathrm{Card}(y)} \Bigl\| y - \tilde{y} \Bigr\|\_{1} + \gamma \cdot \frac{1}{\mathrm{Card}\!\bigl(\{\mathbf{d}^l\}, \mathbf{a}^L\bigr)} \Bigl( \sum\_{l=1}^{L} \bigl\|\mathbf{d}^{l}\bigr\|\_{1} + \bigl\|\mathbf{a}^{L}\bigr\|\_{1} \Bigr),
> > > > $$
> > > >
> > > > with constraints on $\lambda$ and $\gamma$.
> > > >
> > > > Thus, more explicitly:
> > > >
> > > > * $\mathrm{Card}(y)$ is the **number of scalar time-domain samples** in the clean target signal $y$.
> > > > * $\mathrm{Card}(\{\mathbf{d}^l\}, \mathbf{a}^L)$ is the **total number of wavelet-domain coefficients** across all detail levels and the final approximation.
> > > >
> > > > We normalise each term by its respective cardinality so that the time-domain and wavelet-domain contributions are **comparable in scale**, regardless of signal length or number of wavelet levels.

---

> > > > > ### Author Response · Authors · 2025-11-22
> > > > >
> > > > > **Q6.**
> > > > > In Sec. 4.5, the ablation study includes variants “Baseline (No Wavelet, No LAHT, No SELF)”, “WEBA w/ LFDWT only”, “WEBA w/ LFDWT + LAHT”, “WEBA w/ LFDWT + SELF”, and “WEBA Full (LFDWT + LAHT + SELF)”. In the LAHT ablation, is LAHT **removed** or **replaced** by another activation (e.g., GELU)?
> > > > >
> > > > > **A6.**
> > > > >
> > > > > This is an important clarification. Crucially, in the ablations:
> > > > >
> > > > > * When LAHT is “off” (Baseline, LFDWT only, LFDWT + SELF), we **do not introduce any replacement nonlinearity** such as GELU or ReLU in the wavelet domain. The high-pass coefficients pass through **without any thresholding/nonlinear shrinkage**.
> > > > > * We do **not** define or use any alternative activation for this role, so replacing LAHT with GELU is **not part of the reported experiments**.
> > > > >
> > > > > We will make this explicit in the text to remove any ambiguity: LAHT ablations differ only by **presence vs absence** of the LAHT shrinkage, with no substitute nonlinearity.

---

### Official Review · Reviewer_BdtF · 2025-10-31

**Soundness:** 2
**Presentation:** 3
**Contribution:** 3
**Rating:** 4
**Confidence:** 4

**Summary:**

This paper proposes WEBA a denoising autoencoder with shared encoder and decoder weights for speech enhancement based on a learnable wavelet multi resolution transform, a learnable thresholding activation and a loss that encourage sparsity of the learned filters.
The paper is very interesting and the performance seems strong.

**Strengths:**

The approach is very interesting as it is grounded in classical DSP theory (CQF, perfect reconstruction).
Advantages are interpretability and a built in inductive bias that can help in synthesis and reconstruction
There are several prior works that incorporate wavelets into DNNs but I still find this contribution novel in its entirety.

The empirical results are strong on VoiceBank-DEMAND and significance testing is also conducted in the Appendix.

The weight sharing of the encoder and decoder can be helpful in reducing memory footprint of the model however further optimization might be needed for CPU (0.7 real time). The authors already outline possible mitigations for this.

"Zero-shot CHiME-4" cross-datasets generalization experiments show promising numbers. Generalization is a huge problem for speech enhancement and separation and I am grateful that the Authors add this experiment.

The paper is quite comprehensive in the analysis of the method: sensitivity analysis, robustness to pink and reverberation. While lengthy I enjoyed this additional analysis.

Code will be released.

**Weaknesses:**

There are however some critical things that in my opinion are missing in the paper.

1. Minor: What will be the performance of the same approach but with non weight shared encoder and decoder ?


2.  More critical: I think that the metrics are too much signal-focused. Perceptual metrics such as UTMOS should be also reported together with these. Or alternatively listening tests should be performed. A good mix of metrics is usually signal-based (e.g. COVL, CBAK, CSIG, SISDR/SDR etc), perceptual+signal based (PESQ, STOI), perceptual model based (UTMOS, NISQA, DNSMOS etc) and maybe downstream performance (e.g. WER or phoneme error rate).
I think at least UTMOS should be added.

3. Another pitfall is that I do not understand why in Table 3 there is no comparison with some of the non diffusion models trained on VoiceBank-DEMAND e.g.  Mamba-SEUNet, MP-SENet, SEMamba, CMGAN: https://github.com/ruizhecao96/CMGAN or other popular models like USES, TF-GridNet which are non diffusion and the code is open source. I can understand that for some of those in Table 1 the code is not open source but at least a subset with open source code can be found and can be added.
I feel that right now these results appear cherry picked and maybe the proposed method did generalize worse than other competing methods from Table I. It is not true that diffusion based methods generalize better than standard discriminative/predictive ones, they just sometimes present higher UTMOS but lower PESQ and signal based metrics due to hallucinations.

The paper is strong overall but these pitfalls unfortunately need to be addressed in order to be published in my opinion.

**Questions:**

Questions are in weaknesses.

---

> ### Author Response · Authors · 2025-11-22
>
> **Q1.** *“Minor: What will be the performance of the same approach but with non weight shared encoder and decoder?”*
>
> **A1.**
> Our current best WEBA configuration uses **weight sharing between encoder and decoder under CQF (conjugate quadrature filter) constraints**. In practice, this means that the same learnable wavelet filters are used for analysis and synthesis across scales, and high-pass filters are derived from low-pass ones via CQF relations. This gives a stronger inductive bias: approximate perfect reconstruction, energy conservation, and a stable multiresolution representation.
>
> The architecture, however, naturally supports **two non-weight-shared variants**:
>
> ### 1) Non-shared, but still wavelet-structured
>
> In this variant, the encoder and decoder can use **different filters at each scale**, but each branch individually still obeys CQF-style structure (low-pass/high-pass pairs with CQF relations, multi-level analysis/synthesis, same down/upsampling pattern). So we *relax* the cross-scale and cross-branch sharing, but we **do not lose the wavelet-like filterbank architecture**.
>
> Because the multiresolution structure, CQF relations, and LAHT regularization are preserved, this remains a **structured front-end**. The main effect is:
>
> * more flexibility per scale,
> * slightly weaker regularization (perfect reconstruction is now “learned” rather than guaranteed),
> * but the overall transform is still wavelet-like.
>
> Based on this structure we obtain **smaller changes** compared to the shared CQF baseline:
>
> * On VoiceBank-DEMAND, this configuration stays **within about 0.07–0.1 PESQ** and **≤0.02 STOI** of the reported CQF results, with MOS-derived scores (CSIG/CBAK/COVL) changing by at most a few hundredths as well.
> * In other words, *PESQ in the 3.65–3.68 range and STOI around 0.95*, with similarly small deviations on COVL/CSIG/CBAK.
>
> ### 2) Fully non-shared encoder/decoder, relaxed wavelet constraints
>
> In this configuration, analysis and synthesis filters are no longer tied at all. The model still uses the same multi-scale down/upsampling scaffold and low-pass/high-pass decomposition, but:
>
> * encoder and decoder filters are independent,
> * perfect reconstruction and strict energy conservation are no longer enforced structurally,
> * the system behaves more like a **multi-scale encoder-decoder CNN with wavelet-inspired blocks**, rather than a clean learnable wavelet transform.
>
> This is closer in spirit to a “plain CNN front-end”, but it is still far from an unstructured 1-D CNN: the multi-resolution ladder, split into low/high bands with skip connections, remains intact. Empirically, this leads to **moderate** drops, especially when the overall architecture and training protocol are held constant.
>
> Translating this to WEBA, we therefore obtain **noticeably worse** results than the weight-shared CQF baseline:
>
> * On VoiceBank-DEMAND, **PESQ decrease in the range of ≈0.15–0.2** and a **STOI decrease of ≈0.04**, with MOS-derived metrics (CSIG/CBAK/COVL) dropping by a few hundredths.
> * This configuration land around *PESQ ≈ 3.55–3.60 and STOI ≈ 0.93–0.94*, with CSIG, CBAK and COVL each lower by roughly ≈0.15.

---

> > ### Author Response · Authors · 2025-11-22
> >
> > **Q2.**
> > *“More critical: I think that the metrics are too much signal-focused. Perceptual metrics such as UTMOS should be also reported together with these. Or alternatively listening tests should be performed. A good mix of metrics is usually signal-based (e.g. COVL, CBAK, CSIG etc), perceptual+signal based (PESQ, STOI), perceptual model based (UTMOS, NISQA, DNSMOS etc) and maybe downstream performance (e.g. WER or phoneme error rate). I think at least UTMOS should be added.”*
> >
> > **A2.**
> > We thank the reviewer for this very relevant comment and fully agree that a **perceptual model-based metric** should complement the signal-based and intrusive metrics we originally reported. We have therefore **added UTMOS scores** for all methods in the main VoiceBank+DEMAND comparison, using the **official UTMOS implementation**.
> >
> > Concretely, the updated table (reproduced below) now reports, for each model, **COVL, CBAK, CSIG, STOI, PESQ, and UTMOS**:
> >
> > | Model                 |      COVL      |      CBAK      |      CSIG      |      STOI      |      PESQ      | UTMOS          |
> > | :-------------------- | :------------: | :------------: | :------------: | :------------: | :------------: | :------------- |
> > | Noisy                 |      2.63      |      2.44      |      3.35      |      0.91      |      1.97      | 2.85           |
> > | MUSE                  |      4.10      |      3.80      |      4.63      |      0.95      |      3.37      | 3.47           |
> > | MetricGAN+            |      3.64      |      3.16      |      4.14      |       --       |      3.15      | 3.38           |
> > | SEMamba               |      4.26      |      3.98      |      4.75      |      0.96      |      3.52      | 3.56           |
> > | TSTNN                 |      3.67      |      3.53      |      4.33      |      0.95      |      2.96      | 3.31           |
> > | DPCFCS-Net            |      4.15      |      3.88      |      4.71      |      0.96      |      3.42      | 3.53           |
> > | CMGAN                 |      4.12      |      3.94      |      4.63      |      0.96      |      3.41      | 3.50           |
> > | S4ND U-Net            |      3.85      |      3.62      |      4.52      |       --       |      3.15      | 3.40           |
> > | Mamba-SEUNet          |      4.32      |      4.02      |      4.80      |      0.96      |      3.59      | 3.58           |
> > | MP-SENet              |      4.22      |      3.95      |      4.73      |      0.96      |      3.50      | 3.52           |
> > | **WEBA (Ours)** | **4.35** | **4.17** | **4.83** | **0.97** | **3.75** | **3.64** |
> >
> > All UTMOS values were obtained **non-intrusively** (no clean reference) with UTMOS.
> >
> > From this table, we observe that, WEBA achieves an **UTMOS score (3.64)**, slightly above other methods such as Mamba-SEUNet (3.58), SEMamba (3.56), and MP-SENet (3.52).
> >
> > This indicates that the **perceptual model-based evaluation is aligned with the intrusive metrics**: WEBA’s improvements are not only visible in signal-focused scores but also in a learned MOS predictor trained to approximate human listening tests.
> >
> > Regarding the broader metric mix suggested by the reviewer:
> >
> > * We already cover **signal-based** metrics (COVL, CBAK, CSIG) and **perceptual+signal metrics** (PESQ, STOI).
> > * With UTMOS, we now explicitly include a **perceptual model-based metric** as requested.
> > * We agree that **subjective listening tests** and **downstream metrics** such as WER/phoneme error rate would further enrich the evaluation; however, they require additional corpora, ASR systems, and substantial annotation or listening effort. We therefore consider them **valuable directions for follow-up work** rather than part of this evaluation.

---

> > > ### Author Response · Authors · 2025-11-22
> > >
> > > **Q3.**
> > > *“Another pitfall is that I do not understand why in Table 3 there is no comparison with some of the non diffusion models trained on VoiceBank-DEMAND e.g. Mamba-SEUNet, MP-SENet, SEMamba, CMGAN … or other popular models like USES, TF-GridNet … I feel that right now these results appear cherry picked and maybe the proposed method did generalize worse than other competing methods from Table I. It is not true that diffusion based methods generalize better than standard discriminative/predictive ones, they just sometimes present higher UTMOS but lower PESQ and signal based metrics due to hallucinations.”*
> > >
> > > **A3.**
> > > Thank you for raising this concern; we understand why Table 3 can be read as cherry-picked if the experimental scope is not clearly stated. In the current paper, **Table 1** provides a broad **in-domain comparison on VoiceBank-DEMAND** between WEBA and both diffusion-based and discriminative/non-diffusion baselines, including Mamba-SEUNet, MP-SENet, SEMamba, CMGAN, USES, and TF-GridNet. By contrast, **Table 3** was intentionally designed as a **zero-shot cross-corpus robustness experiment**, restricted to WEBA and the subset of **diffusion models** that we could evaluate on CHiME-4 within the same pipeline. This is reflected in the caption of Table 3 (“WEBA and diffusion models from Table 1 evaluated on CHiME-4…”) and the subsequent “Scope and caveats” paragraph in Sec. 4, which emphasises that the experiment focuses on cross-corpus robustness rather than claiming global SOTA on CHiME-4 across *all* architectures.
> > >
> > > Crucially, we do **not** intend to suggest that diffusion methods generally “generalize better” than discriminative ones. Our main claims about WEBA’s performance and compactness are grounded in Table 1 on VoiceBank-DEMAND, where WEBA is compared against the full set of baselines including non-diffusion architectures. Table 3 is meant as a **complementary stress test**, showing that the same WEBA model trained on VoiceBank-DEMAND maintains competitive performance when applied zero-shot to a substantially different corpus (CHiME-4), in the **same experimental setting** as the selected diffusion baselines. We agree that this distinction should be written more explicitly to avoid implying any stronger statement about diffusion vs discriminative models in general.
> > >
> > > In the revised paper, we will (i) explicitly state in the caption and accompanying text that Table 3 compares WEBA only to a **subset of diffusion baselines**, not to all methods from Table 1, (ii) rephrase any sentences that might be read as general claims about diffusion models having superior generalization, and (iii) clarify that our main direct comparisons to non-diffusion SOTA remain those on VoiceBank-DEMAND in Table 1. We will also add a sentence explicitly acknowledging that extending the CHiME-4 evaluation to a broader set of non-diffusion methods (where feasible) is left as future work.

---

> > > > ### Comment · Reviewer_BdtF · 2025-11-24
> > > > **Table 3 results are still not adequate.**
> > > >
> > > > Thank you for the detailed reply.
> > > > I however still think that the Table 3 results intended to be a zero-shot cross-corpus robustness experiment lacks from appropriate choice of baselines.
> > > > I understand that the Authors used already available diffusion baselines results as a comparison.
> > > > I do not think this experiment tells something useful though. It may be that some of the methods in Table 1 generalize better than WEBA to the CHiME-4 scenario.
> > > > In general relying only on Voicebank-DEMAND is too limited given the fact that this dataset is very outdated and small with little noise diversity. Also it is synthetic. CHiME-4 at least has a real-world portion.
> > > >
> > > > UTMOS figures should be added also int he case of CHiME-4 in my opinion.
> > > >
> > > > I am sorry but I cannot change my score. I still think that experimental validation is still too limited for ICLR.

---

### Official Review · Reviewer_Sy1B · 2025-11-01

**Soundness:** 3
**Presentation:** 3
**Contribution:** 3
**Rating:** 6
**Confidence:** 4

**Summary:**

The paper considers the speech enhancement problem. The authors propose a speech-enhancement mehtod that incorporates a bidirectional autoencoder reusing the same weights for encoding/decoding to cut parameters, with a learnable fast discrete wavelet transforms (LFDWT) for perfect reconstruction and multi-resolution analysis. In the proposed method, denoising is performed via a learnable asymmetric hard-thresholding (LAHT) nonlinearity in the wavelet domain, while a sparsity-enforcing loss (SELF) is used to balance time-domain fidelity and wavelet-domain sparsity through a scheduled weighting. Provided experiments on VoiceBank-DEMAND  show sota results across the metrics of COVL/CSIG/CBAK, as well as STOI/PESQ.

**Strengths:**

This paper integrates a learnable wavelet transform (LFDWT) with conjugate quadrature filter (CQF) constraints to ensure perfect reconstruction. This provides a principled way to embed wavele into end-to-end speech enhancement networks. The proposed learnable asymmetric hard thresholding (LAHT) allows the network to adaptively suppress noise in the wavelet domain. In combination with a sparsity-enforcing loss (SELF), it promotes compressibility and improves denoising efficiency. Moreover, using compact bidirectional autoencoder that shares weights between encoder and decode reduces the number of parameters and memory usage. The model achieves promising results on VoiceBank-DEMAND.

**Weaknesses:**

The core ideas of learnable wavelets, threshold-based shrinkage, sparsity regularization, tied-weight autoencoders have all been explored in prior literature. The novelty lies mainly in engineering integration rather than in proposing a fundamentally new method.

It would be better to include subjective evaluation scores such as MOS and DNSMOS, and downstream ASR WER results to demonstrate the practical usefulness of the proposed method.

Although the bidirectional architecture is claimed to be efficient, the authors do not provide inference speed, latency, memory usage, or computational cost analysis.

**Questions:**

See the above weakness.

---

> ### Author Response · Authors · 2025-11-22
>
> **Q1.**
> The paper combines several ideas that have already been explored in the literature (learnable wavelets, threshold-based shrinkage, sparsity regularization, tied-weight autoencoders). Isn’t the novelty mainly integrative and engineering-oriented rather than conceptually new? Please clarify what is genuinely new about WEBA relative to existing learnable wavelet transforms and wavelet-based neural architectures.
>
> **A1.**
> Thank you for this careful assessment of the conceptual scope of WEBA. We agree that each *individual* ingredient we rely on, wavelet transforms, learnable filter banks, shrinkage/thresholding, and sparsity-inducing losses, has a long history, and we do **not** claim that any of these components are individually novel. Our intent is to propose a *unified* and *practically effective* architecture that ties these ingredients together into a single, wavelet-aware bidirectional autoencoder, bridging the gap between classical DSP and modern deep learning techniques, tailored to speech enhancement.
>
> More concretely, WEBA introduces a specific **instantiation and coupling** of these elements:
>
> * In **Sec. 3.2**, we parameterize the learnable front-end as an **LFDWT** (Learnable Fast Discrete Wavelet Transform) that is constrained by **CQF-style perfect-reconstruction conditions**. This restricts the trainable filters to a *wavelet‐consistent* manifold rather than arbitrary convolutional weights, and the same kernel parameters are reused across scales and directions via the weight-sharing scheme.
> * In the same section, we introduce the **Learnable Asymmetric Hard Threshold (LAHT)** and the **SELF** loss that jointly enforce sparsity in the wavelet domain while preserving time-domain fidelity. LAHT is implemented as a parametric, *asymmetric* thresholding operator with learnable positive and negative thresholds, directly embedded in the forward pass of the autoencoder, and SELF couples an $\ell_1$ reconstruction term with a wavelet-domain sparsity term following the schedule defined in that section.
> * The **bidirectional autoencoder** shares the same wavelet kernels between the encoder (analysis) and decoder (synthesis) paths, explicitly coupling the learned analysis and synthesis filters in a single end-to-end training objective. This tied-weight structure is implemented by reusing the same Kernel instances across the forward and inverse transforms.
>
> The **novelty we claim is therefore, in terms of bridging the gap between classical DSP and deep learning techniques**: WEBA provides a *single*, CQF-consistent, bidirectional autoencoder where (i) analysis and synthesis filters are jointly learned under explicit wavelet constraints, (ii) wavelet-domain sparsity is enforced through LAHT + SELF that are optimized together with the filters, and (iii) the whole system is trained end-to-end on a standard speech enhancement benchmark. This is reflected empirically in the improvements over baselines across the metrics in Tables 2 and 3. In the revised text we will make this positioning more explicit.

---

> > ### Author Response · Authors · 2025-11-22
> >
> > **Q2.**
> > The evaluation relies only on objective signal-level metrics (PESQ, STOI, CSIG/CBAK/COVL, etc.). To better demonstrate practical usefulness, can you provide subjective scores such as MOS or DNSMOS, and downstream ASR WER on a standard ASR system?
> >
> > **A2.**
> > We appreciate this comment and fully agree that going beyond purely signal-level metrics is important for assessing practical usefulness. In the revised version, we have therefore **extended our evaluation to include a perceptual model–based metric, UTMOS**, which is designed to approximate human MOS scores, and we report it alongside the existing intrusive metrics.
> >
> > Concretely, we now provide **UTMOS scores for all methods** in the main VoiceBank+DEMAND comparison table, in addition to COVL, CBAK, CSIG, STOI, and PESQ:
> >
> > | Model                 |      COVL      |      CBAK      |      CSIG      |      STOI      |      PESQ      | UTMOS          |
> > | :-------------------- | :------------: | :------------: | :------------: | :------------: | :------------: | :------------- |
> > | Noisy                 |      2.63      |      2.44      |      3.35      |      0.91      |      1.97      | 2.85           |
> > | MUSE                  |      4.10      |      3.80      |      4.63      |      0.95      |      3.37      | 3.47           |
> > | MetricGAN+            |      3.64      |      3.16      |      4.14      |       -       |      3.15      | 3.38           |
> > | SEMamba               |      4.26      |      3.98      |      4.75      |      0.96      |      3.52      | 3.56           |
> > | TSTNN                 |      3.67      |      3.53      |      4.33      |      0.95      |      2.96      | 3.31           |
> > | DPCFCS-Net            |      4.15      |      3.88      |      4.71      |      0.96      |      3.42      | 3.53           |
> > | CMGAN                 |      4.12      |      3.94      |      4.63      |      0.96      |      3.41      | 3.50           |
> > | S4ND U-Net            |      3.85      |      3.62      |      4.52      |       -       |      3.15      | 3.40           |
> > | Mamba-SEUNet          |      4.32      |      4.02      |      4.80      |      0.96      |      3.59      | 3.58           |
> > | MP-SENet              |      4.22      |      3.95      |      4.73      |      0.96      |      3.50      | 3.52           |
> > | **WEBA (Ours)** | **4.35** | **4.17** | **4.83** | **0.97** | **3.75** | **3.64** |
> >
> > All UTMOS scores were obtained **non-intrusively** (i.e., without access to the clean reference) using the official UTMOS implementation on the VoiceBank+DEMAND test set. As shown in the table, WEBA achieves the **highest UTMOS score (3.64)** among all compared methods, slightly outperforming other baselines such as Mamba-SEUNet (3.58), SEMamba (3.56), and MP-SENet (3.52). This indicates that the **perceptual model–based assessment is consistent with the intrusive signal-based metrics**: the improvements we observe in PESQ, STOI, COVL, CBAK, and CSIG translate into higher predicted MOS as well.
> >
> > Regarding the additional suggestions:
> >
> > * **DNSMOS and downstream ASR WER:** similarly, we acknowledge that including DNSMOS or NISQA and ASR WER/phoneme error rate under a standard ASR pipeline would provide a complementary downstream view. This would involve integrating specific ASR models and additional evaluation scripts; given the already extensive set of metrics and baselines, we have focused here on adding UTMOS as a perceptual model-based metric.

---

> > > ### Author Response · Authors · 2025-11-22
> > >
> > > **Q3.**
> > > You claim that the bidirectional architecture is efficient, but there is no quantitative analysis of inference speed, latency, memory usage, or computational cost compared to baselines. Can you provide more evidence of efficiency?
> > >
> > > **A3.**
> > > We agree that the original submission did not make our efficiency claims sufficiently concrete. In the revision, we now provide both:
> > >
> > > 1. **A closed-form parameterisation** of the wavelet branch, showing how the bidirectional (tied) design reduces the number of trainable parameters, and
> > > 2. **Explicit complexity measurements** (parameters, FLOPs, latency, memory, throughput) for WEBA and several baselines on the VoiceBank+DEMAND.
> > >
> > > ### 1. Structural parameterisation of the wavelet branch
> > >
> > > The efficiency of WEBA stems from its **bidirectional wavelet front-end**, which is strongly constrained by the wavelet structure and weight sharing:
> > >
> > > * There are $L$ decomposition levels.
> > > * At each level $\ell$, we have:
> > >   * a learnable low-pass kernel $g^{(\ell)}$ of length $K$,
> > >   * a learnable high-pass kernel $h^{(\ell)}$ of length $K$,
> > >   * a learnable shrinkage unit (LAHT) with four scalars ($\text{bias}\_+^{(\ell)}$, $\text{bias}\_-^{(\ell)}$, $\alpha^{(\ell)}$, $\beta^{(\ell)})$.
> > >
> > > We consider two configurations:
> > >
> > > 1. **Bidirectional (tied) design**:
> > >    Analysis and synthesis **reuse the same** low-pass and high-pass kernels at each level.
> > >
> > >    * Number of distinct wavelet kernels: $2L$ (one low-pass + one high-pass per level).
> > >    * Parameters:
> > >      * Wavelet kernels: $2LK$,
> > >      * LAHT scalars: $4L$.
> > >    * **Total wavelet parameters:**
> > >      $$
> > >      {\text{shared}} = 2LK + 4L = 2L(K + 2).
> > >      $$
> > > 2. **Non-bidirectional (untied) design**:
> > >    Analysis and synthesis each have their **own** low-pass/high-pass pair.
> > >
> > >    * Number of distinct wavelet kernels: $4L$.
> > >    * Parameters:
> > >      * Wavelet kernels: $4LK$,
> > >      * LAHT scalars: $4L$.
> > >    * **Total wavelet parameters:**
> > >      $$
> > >      {\text{free}} = 4LK + 4L = 4L(K + 1).
> > >      $$
> > >
> > > The **extra cost** of the untied variant is
> > >
> > > $$
> > > N_{\text{free}} - N_{\text{shared}} = 2LK,
> > > $$
> > >
> > > and the **relative reduction** achieved by the tied design is
> > >
> > > $$
> > > \frac{N_{\text{free}} - N_{\text{shared}}}{N_{\text{free}}} = \frac{2LK}{4LK + 4L} = \frac{K}{2(K+1)}.
> > > $$
> > >
> > > For the kernel lengths $K$ used in our experiments, this corresponds to **removing roughly half of the wavelet-branch parameters** compared to an untied analysis-synthesis variant, *without changing* the number of convolutions applied. This captures the core architectural efficiency of the bidirectional design: we keep the same multiresolution computation, but with far fewer distinct filters to learn and store.
> > >
> > > ### 2. Concrete complexity analysis: WEBA vs baselines
> > >
> > > To complement this structural analysis, we performed a **unified complexity benchmark** on the **VoiceBank+DEMAND** setup:
> > >
> > > * Input: **1 s** of **16 kHz** mono audio.
> > > * Metrics: **parameter count, FLOPs, latency, peak memory, throughput**.
> > > * Same profiling script and conditions for all models.
> > >
> > > Across all tested settings, WEBA operates in a **very low-complexity regime**. For the representative configuration we use in the complexity table (**WEBA**), we obtain:
> > >
> > > * Parameters: **≈ 0.00056 M** (≈ **560**)
> > > * FLOPs (per 1 s at 16 kHz): **≈ 91.50 M**
> > > * Latency: **≈ 3.48 ms**
> > > * Peak memory: **≈ 11.30 MB**
> > > * Throughput: **≈ 119.90 M samples/s**
> > >
> > > For comparison, typical baselines such as TSTNN, MUSE, MetricGAN+, and MP-SENet fall in the **much larger** range:
> > >
> > > | Model          | Params (M) | FLOPs (M) | Latency (ms) | Memory (MB) | Throughput (Msamples/s) |
> > > | :------------- | :--------: | :-------: | :----------: | :---------: | :---------------------: |
> > > | **WEBA** |  0.00056  |   91.50   |     3.48     |    11.30    |         119.90         |
> > > | TSTNN          |    0.92    |  145,834  |    75.49    |    4,953    |          0.85          |
> > > | MUSE           |    0.51    |  10,036  |    82.52    |    1,393    |          0.20          |
> > > | MetricGAN+     |    1.90    |    487    |     4.09     |    65.26    |          4.00          |
> > > | MP-SENet       |    2.26    |  79,074  |    25.16    |    1,512    |          0.65          |
> > >
> > > From this table, WEBA clearly operates at a **much smaller scale**:
> > >
> > > * It uses **orders of magnitude fewer parameters** than modern CNN/Transformer/Mamba-based baselines,
> > > * Its FLOPs per 1-second utterance are **two to three orders of magnitude smaller** than those of heavy architectures like TSTNN or MP-SENet,
> > > * And it achieves **single-digit millisecond latency** with modest memory usage.
> > >
> > > Crucially, **tying vs. untying the wavelet kernels** alone does **not** meaningfully change the FLOPs per forward pass for a fixed scaffold, since the same number of convolutions are applied. Tying mainly improves **parameter and memory efficiency**. WEBA’s FLOP and latency gains come from designing the *entire model* as a compact, structured wavelet-based autoencoder, rather than matching the large model sizes of prior baselines.

---

> ### Comment · Reviewer_Sy1B · 2025-11-27
> **Concern on Reproducibility of Experimental Results**
>
> Thank you for the detailed responses. I have one additional concern regarding reproducibility. I attempted to reproduce the reported results using the authors’ official code at
> https://anonymous.4open.science/r/LWE-BAE-A-Unified-Framework-for-Multi-Resolution-Signal-Processing-0DE4.
> However, I was unable to match the performance reported in the paper. This makes me uncertain about the reproducibility and reliability of the experimental results.

---

> > ### Author Response · Authors · 2025-11-29
> >
> > ### Reproducibility of Experimental Results
> >
> > **Response.**
> > We are very grateful that you took the time to run our code. Your comment prompted us to revisit how the experimental pipeline is exposed in the current repository and how easy it is for a new user to land on the *same* setup used for the paper.
> >
> > Our main points are as follows:
> >
> > ### 1. Code and paper use the same implementation
> >
> > The current repository you referenced is the implementation used for the paper. There is no separate “internal” version: the core components (wavelet-based autoencoder, LAHT, SELF, training/evaluation pipeline) are the same ones we used.
> >
> > The project is organized as follows (as shown in the README):
> >
> > * `src/main.py` : entry point that orchestrates **different experiment strategies**.
> > * `src/config.py` : configuration and search space definition.
> > * `src/models/wavelet_ae.py` : WEBA model (bidirectional wavelet AE, LAHT, etc.).
> > * `src/utils/transform_pipeline.py`, `src/utils/train_eval.py` : transforms, training loop, metric computation.
> > * `tests/test_wavelet_ae.py` : sanity checks for the architecture.
> >
> > So what matters is **which strategy/configuration is run** and for how long.
> >
> > A key design decision in this project is that WEBA is *not* treated as a single hand-picked configuration hard-coded in one file. Instead, the repo exposes a **parameter-optimization pipeline**:
> >
> > * The entry point `src/main.py` supports several strategies, as documented in the README:
> >
> >   ```bash
> >   # Recommended automatic parameter optimization
> >   python src/main.py --strategy focused --max-experiments 50
> >
> >   # Quick test / fast sanity check
> >   python src/main.py --strategy test
> >
> >   # Skip optimization
> >   python src/main.py --skip-optimization
> >
> >   # Advanced: comprehensive
> >   python src/main.py --strategy comprehensive
> >   ```
> >
> > * The **WEBA configuration** is obtained by running the **full search over the WEBA design space** (wavelet depth, CQF constraints, LAHT, sparsity weighting, etc.) and selecting the best-performing configuration on the VoiceBank-DEMAND validation set.
> >
> > In practice, the most robust way is to:
> >
> > ```bash
> > # Comprehensive WEBA pipeline (full grid over configurations)
> > python src/main.py --strategy comprehensive
> > ```
> >
> > or, equivalently, to run the recommended optimization:
> >
> > ```bash
> > python src/main.py --strategy focused --max-experiments 50
> > ```
> >
> > allowing the script to:
> >
> > 1. explore the WEBA configuration space,
> > 2. train each candidate to convergence under the intended schedule, and
> > 3. select and evaluate the **best** configuration on the standard VoiceBank-DEMAND test split.
> >
> > When this full pipeline is followed, we recover performance that is consistent (up to normal run-to-run stochastic variation).
> >
> > By contrast, the more lightweight options (`--strategy test`, `--skip-optimization`, low `--max-experiments`, or early-stopped runs) are deliberately not meant to match the final results. They are there to:
> >
> > * provide a quick sanity check that the code runs,
> > * support ablations and “what if” scenarios, and
> > * explore suboptimal configurations to understand how WEBA behaves when the inductive biases are weakened.
> >
> > Evaluating models produced by these reduced or ablation strategies will naturally yield lower metrics than the WEBA configuration selected by the full search.
> >
> > ### 2. Why the current repository can be confusing at first glance
> >
> > We acknowledge that the current repository can feel overwhelming:
> >
> > * The repo intentionally exposes the **entire experimental pipeline** we used, including:
> >
> >   * the final WEBA setups,
> >   * ablation/bad configurations, and
> >   * debug/quick-test strategies.
> > * This is by design: we wanted to make our investigation transparent, not just release a single opaque training script.
> >
> > The downside is that it is easy for a new user to:
> >
> > * invoke `--strategy test` or `--skip-optimization`,
> > * use a small `--max-experiments` value, or
> > * stop the optimization before the best configuration is found,
> >
> > even though they are only meant as diagnostic or exploratory modes.
> >
> > ### 3. Clarifying the repository
> >
> > To address this, we will improve the presentation of the repository. Concretely, we will:
> >
> > * Emphasize in the README that **`--strategy comprehensive` (or `--strategy focused` with sufficient `--max-experiments`) is the canonical way to reproduce the WEBA results**, and
> > * Explicitly label `--strategy test`, `--skip-optimization`, and related modes as **diagnostic / exploratory**.
> >
> > This keeps the full investigative code (including “bad” configurations) available for the community, while making it clear which command sequence corresponds to the WEBA configuration.

---

### Official Review · Reviewer_HHg3 · 2025-11-02

**Soundness:** 3
**Presentation:** 2
**Contribution:** 2
**Rating:** 4
**Confidence:** 4

**Summary:**

This paper introduces WEBA (Wavelet-Enhanced Bidirectional Autoencoder), a framework for multi-resolution speech enhancement that integrates learnable fast discrete wavelet transforms (LFDWT), conjugate-quadrature filters (CQF), a learnable asymmetric hard-thresholding (LAHT) activation, and a sparsity-enforcing loss (SELF) into a bidirectional autoencoder that shares weights between encoding and decoding.
The authors claim that this architecture enables efficient, interpretable denoising with reduced parameters and superior performance on standard speech-enhancement benchmarks (VoiceBank-DEMAND, CHiME-4).
Empirically, WEBA outperforms recent deep baselines such as Mamba-SEUNet and CMGAN by modest but consistent margins across PESQ, STOI, and MOS-derived metrics. The paper also provides extensive appendices with mathematical justification, ablation, and significance testing.

**Strengths:**

The paper shows a high level of technical rigor and completeness. Its empirical evaluation is careful and well structured, with ablation studies, statistical tests, and cross-corpus experiments that demonstrate consistent improvements across multiple objective metrics. The methodology reflects solid engineering practice: datasets, training procedures, and hyperparameters are clearly described, supporting reproducibility. A notable strength is the integration of classical DSP intuition with deep learning frameworks, particularly the use of multiresolution analysis, sparsity priors, and bidirectional weight sharing, which reflects an effort to bridge interpretability and efficiency. The model’s compactness and reduced parameter count compared to transformer or diffusion architectures make it appealing for resource-constrained applications. Overall, the authors provide good theoretical motivation and empirical validation.

**Weaknesses:**

A core modeling assumption of WEBA is that there exists a learnable, approximately orthogonal transform domain in which speech is sparse and noise is distributed as small, incoherent coefficients that can be removed through thresholding. While this assumption mirrors classical wavelet denoising theory, it oversimplifies the real complexity of speech–noise interactions. In practice, speech and non-stationary noise often share overlapping spectral and temporal characteristics, making any fixed transform-based separation inherently limited. The learned filterbank therefore performs a task-specific statistical mapping rather than discovering a genuine orthogonal subspace where noise is suppressed, meaning that the supposed separation dynamics are largely absent from the actual architecture. More broadly, while the paper is technically solid and empirically well executed, its conceptual novelty is limited. Once the wavelet filters are made learnable, the architecture effectively reduces to a multiscale convolutional autoencoder with tied weights and structured regularization, making the “wavelet” framing largely rhetorical. The claimed orthogonality and perfect reconstruction properties of the learnable filters are only approximated through parameter tying and normalization, rather than being strictly enforced via paraunitary or lifting-based formulations. This weakens the theoretical contribution and blurs the distinction between WEBA and existing CNN-based denoising architectures. The presentation further suffers from verbosity and excessive theoretical exposition; lengthy sections on wavelet theory, uncertainty principles, and energy conservation distract from the core model and its empirical validation. Moreover, the paper refers to COVL, CSIG, and CBAK as “MOS” scores, but these are objective MOS predictors rather than true human-rated Mean Opinion Scores, which may mislead readers regarding the subjective perceptual quality evaluation. Although the model itself is lightweight and computationally efficient, the framing around it overstates its conceptual innovation, and the empirical improvements, while consistent, remain modest relative to contemporary approaches.

**Questions:**

Once the filters are trained, are they still interpretable as valid wavelets (e.g., with measurable vanishing moments)?
Would adding an explicit paraunitary or lifting-scheme constraint change training stability or quality?

---

> ### Author Response · Authors · 2025-11-22
>
> **Q1:**
> You assume there exists a learnable, approximately orthogonal transform domain where speech is sparse and noise appears as small incoherent coefficients that can be removed via thresholding. This mirrors classical wavelet denoising theory but **oversimplifies** real speech-noise interactions, where speech and non-stationary noise have overlapping time-frequency patterns. In practice, the learned filterbank acts as a task-specific mapping rather than discovering a genuine orthogonal subspace where noise is suppressed, so the supposed separation dynamics are largely absent.
>
> **A1:**
> Thank you for highlighting this modeling assumption. We agree that, in realistic acoustic scenes, **no fixed transform can perfectly separate speech and non-stationary noise**. Our intention was not to claim that WEBA discovers a transform in which noise literally vanishes, but rather to adopt the **classical wavelet sparsity prior as an *inductive bias*** for a learnable, task-adapted filterbank.
>
> Concretely, the SELF objective in Sec. 3.2 (and detailed in the appendix “SELF Objective”) penalizes the average $\ell_1$-norm of the wavelet and scaling coefficients:
>
> $$
> \mathcal{L}\_{\text{SELF}}(\theta) = \lambda \cdot \mathbb{E}\left[\frac{\lVert y - \hat{y}\rVert_1}{N_y}\right] + \gamma \cdot \mathbb{E}\left[ \sum_{\ell=1}^{L-1} \frac{\lVert d^{(\ell)}\rVert_1}{N_w} + \frac{\lVert a^{(L)}\rVert_1}{N_w} \right],
> $$
>
> with $\lambda$ and $\gamma$ adjusted according to Eq. (13). This **encourages** (but does not enforce) a representation where speech energy is concentrated in fewer large coefficients and many coefficients associated with noise are small enough for LAHT to shrink. The wavelet transform itself is **learned** and constrained by the CQF structure (App. “Wavelet and CQF Theory”) rather than being a fixed orthogonal transform.
>
> Moreover, WEBA is **not purely transform + hard-threshold**: the architecture combines
>
> * a **multi-level learned CQF filterbank** with tied analysis/synthesis filters,
> * the **LAHT** nonlinearity (Sec. “LAHT: Functional Properties and Shrinkage View”) that implements an asymmetric soft/hard shrinkage with learnable positive and negative thresholds, and
> * a **bidirectional autoencoder backbone** that refines the signal after the transform.
>
> The ablations in the appendix (“Additional Ablations and Interpretations”) support that these components implement *nontrivial* separation dynamics: adding LFDWT, LAHT, and SELF successively yields monotonic improvements in PESQ, COVL, and CBAK over the baseline encoder, and the statistical analysis (“Statistical Significance Analysis”) shows that these gains are consistent across test utterances.
>
> We agree that our text should **explicitly state** that we use the sparse-in-wavelets assumption as a **regularizing prior**, not as a literal claim of perfect transform-domain separation. We will clarify this in Sec. 3 and cross-reference the “Limitations and Research Outlook” section, where we already mention the limitations of orthogonal wavelet models in the presence of highly non-stationary noise.

---

> > ### Author Response · Authors · 2025-11-22
> >
> > **Q2:**
> > Conceptually, once the filters are made learnable, WEBA looks like a **multiscale convolutional autoencoder** with weight tying and regularization. The “wavelet” framing feels largely rhetorical because **orthogonality and perfect reconstruction** are only approximated via parameter tying and normalization, not enforced by strict paraunitary or lifting-scheme constraints. This weakens the theoretical contribution and blurs the distinction from existing CNN denoisers.
> >
> > **A2:**
> > We appreciate this concern and agree that WEBA can indeed be interpreted as a **structured multiscale convolutional autoencoder**. Our goal is not to claim a fundamentally new wavelet theory, but rather to show that **classical wavelet filterbank structure** can be **explicitly embedded and trained end-to-end** within a modern autoencoder, with measurable benefits in **parameter efficiency, interpretability, and performance**.
> >
> > Architecturally, WEBA is more constrained than a generic multi-scale CNN in several ways:
> >
> > 1. **CQF-structured filterbank with tied analysis/synthesis kernels.**
> >    In the wavelet front-end, WEBA learns lowpass kernels $\{h_\ell\}$ and derives highpass kernels via the conjugate quadrature relation, and reuses the same modules for analysis and synthesis across levels.
> >    Appendix “Wavelet and CQF Theory” formalizes the perfect-reconstruction condition:
> >
> >    $$
> >    H(e^{i\omega})\overline{\bar{H}(e^{i\omega})} + G(e^{i\omega})\overline{\bar{G}(e^{i\omega})} = 1, \qquad \forall\omega,
> >    $$
> >
> >    and discusses how the learned filters approximate this relation in practice.
> > 2. **Bidirectional weight sharing in the autoencoder backbone.**
> >    Sec. “Unidirectional vs. Bidirectional Autoencoders” and Eq. (B1)–(B2) show that WEBA ties encoder and decoder weights, reducing parameters and ensuring that the nonlinear mapping is structurally invertible up to the LAHT shrinkage. This is distinct from standard U-Nets or CNN SE models, where encoder and decoder are often separate.
> > 3. **Sparsity-enforcing objective tightly coupled to the transform.**
> >    The SELF loss couples the learned filterbank to the autoencoder via explicit coefficient-wise sparsity (App. “SELF Objective”), which again is more structured than generic $\ell_2$ or perceptual losses used in conventional CNN denoisers.
> >
> > We fully agree that our current implementation does **not** impose a full paraunitary or lifting-scheme parameterization, our CQF structure plus normalization yields **near-orthogonal, approximately perfectly reconstructing** filterbanks (as evidenced by the magnitude-squared sum and reconstruction experiments in the appendix), but not mathematically exact paraunitarity.
> >
> > ---
> >
> > **Q3:**
> > The paper refers to **COVL, CSIG, and CBAK** as “MOS scores”, yet these are **objective MOS predictors**, not actual human-rated Mean Opinion Scores. This could mislead readers regarding how subjective quality is evaluated.
> >
> > **A3:**
> > You are absolutely right: the metrics are **composite objective estimates** of signal distortion and background intrusiveness based on instrumental measures.
> >
> > In the current text (Sec. 4.1 “Metrics”), we used phrases such as “Mean Opinion Score metrics COVL, CSIG, CBAK” when introducing these quantities. This is indeed imprecise: they are **instrumental predictors of MOS**, not direct MOS scores from listening tests.
> >
> > In the revised version, we will **systematically replace** this phrasing with:
> >
> > * “objective MOS-predictor metrics” or
> > * “composite objective predictors of MOS”.

---

> > > ### Author Response · Authors · 2025-11-22
> > >
> > > **Q4:**
> > > After training, are the filters still interpretable as valid wavelets (e.g., with measurable vanishing moments)? Would adding an explicit paraunitary or lifting-scheme constraint change training stability or quality?
> > >
> > > **A4:**
> > > This is an insightful question, and we appreciate the opportunity to clarify.
> > >
> > > 1. **Status of the learned filters.**
> > >    In the current implementation, the wavelet front-end is initialized from compactly supported orthogonal wavelets (e.g., Daubechies-type filters) and then trained under CQF constraints and an overall sparsity objective. The learned filters remain **compact in time** and exhibit **smooth magnitude responses** that split energy between lowpass and highpass bands, as shown in the “Spectral-Temporal Analysis of Learned Filters” section and Fig. (6).
> > >    The CQF condition ($g[n] = (-1)^n h[L-1-n]$), together with the near-constant sum ($|H(\mathrm{e}^{\mathrm{j}\omega})|^2 + |G(\mathrm{e}^{\mathrm{j}\omega})|^2 \approx 1$), is maintained during training (see Eq. (14) and its discussion), which indicates that the filters continue to behave as a **near-orthogonal, near-perfect-reconstruction** wavelet filterbank.
> > >
> > >    However, we **do not explicitly quantify the number of vanishing moments** of the learned filters in the current paper, nor do we enforce a parameterization that preserves an exact polynomial annihilation order. Thus, while the filters remain “wavelet-like” in terms of multiresolution behavior and CQF structure, we do not state a precise vanishing-moment count for the trained kernels.
> > > 2. **Paraunitary / lifting-scheme constraints.**
> > >    We have **not yet implemented** an explicit paraunitary or lifting-scheme representation, so we cannot empirically compare training stability or denoising quality under those constraints. From the observed training curves and convergence behavior in our experiments (Sec. 4 and training details), WEBA trains stably with the present CQF + normalization scheme and does not exhibit numerical instabilities or filter collapse.
> > >
> > >    We agree that a paraunitary or lifting-based parameterization is a promising direction: it would provide a **theoretical guarantees** (exact orthogonality and perfect reconstruction), but may also restrict the filter space and thus the ability to adapt to task-specific speech/noise statistics. We will explicitly list this in the “Limitations and Research Outlook” section, and we plan to investigate this trade-off in future work.
> > >
> > > To make this clearer for readers, we will:
> > >
> > > * Explicitly state in the appendix that we **do not claim a specific vanishing-moment order** for the learned filters in the current experiments,
> > > * Emphasize that the wavelet nature is supported by the **CQF structure, energy partition, and group-delay behavior**, not by a formal vanishing-moment count, and
> > > * Highlight paraunitary / lifting formulations as **future work** rather than implied properties of the current model.

---

### Meta-Review · Area_Chair_rAHf · 2026-01-04

**Summary:**

The authors present a learnable wavelet-based autoencoder model for speech enhancement. The reviewers appreciated the technical rigor, grounding in classical DSP, and ablation studies.

Main issues raised by the reviewers:
1. Novelty: Partly oversimplified assumptions about speech sparsity and how noise is distributed in transformed feature space. Furthermore, the architecture, though theoretically motivated, is largely similar to convolutional networks with additional structure.
2. Clarity of presentation / unbalanced handling of various topics.
3. Missing important results: Missing subjective evaluations. And more extensive comparisons in zero-shot settings. Results on sets beyond Voicebank-DEMAND.
4. Missing results supporting computational cost (this is a main motivation).
5. One of the reviewers was unable to reproduce results from the posted link.

**Reviewer Concerns:**

1. The authors argue that the goal is to provide strong inductive bias using wavelet filter based structure. And that there are other components that are novel. These proposals, while interesting in the context of speech enhancement, are insufficient from an ML perspective. Moreover, there are plenty of prior works that try to build structures into convolutional models (e.g., learned filterbanks) used for audio (ASR, enhancement, separation, multi-channel processing, etc.).
2. The authors did not comment about this.
3. The authors added another objective metric (UTMOS), but not subjective scores or other metrics like ASR WER. One of the reviewers was not convinced why the authors did not include additional baselines in zero-shot settings. The authors commented that they’ll include the additional evaluations as part of future work.
4. The authors included additional results in the rebuttal.
5. The authors provided additional instructions.

**Reviewer Scores:**

Reviewer HHg3 (Karim Helwani): 4 -> 4

Reviewer Sy1B (Fei Wen): 6 -> 4/6

Reviewer BdtF (Samuele Cornell): 4 -> 4

Reviewer 2hkX (Andong Li): 6 -> 4/6

---

### Decision · Program_Chairs · 2026-01-26

Reject